# NeXT-IMDL: Build Benchmark for NeXT-Generation Image Manipulation Detection & Localization

## Abstract

The accessibility surge and abuse risks of user-friendly image editing models have created an urgent need for generalizable, up-to-date methods for Image Manipulation Detection and Localization (IMDL). Current IMDL research typically uses cross-dataset evaluation, where models trained on one benchmark are tested on others. However, this simplified evaluation approach conceals the fragility of existing methods when handling diverse AI-generated content, leading to misleading impressions of progress. This paper challenges this illusion by proposing NeXT-IMDL, a large-scale diagnostic benchmark designed not just to collect data, but to probe the generalization boundaries of current detectors systematically. Specifically, NeXT-IMDL categorizes AIGC-based manipulations along four fundamental axes: editing models, manipulation types, content semantics, and forgery granularity. Built upon this, NeXT-IMDL implements five rigorous cross-dimension evaluation protocols. Our extensive experiments on 11 representative models reveal a critical insight: while these models perform well in their original settings, they exhibit systemic failures and significant performance degradation when evaluated under our designed protocols that simulate real-world various generalization scenarios. By providing this diagnostic toolkit and the new findings, we aim to advance the development towards building truly robust, next-generation IMDL models.

## 1 Introduction

The rapid popularization of easy-to-use generative image editing models (AI, 2023; Zhao et al., 2024a) has largely reduced the barrier of creating authentic manipulated images, shifting from requiring professional Photoshop expertise to a one-click prompt conversation with Multimodal Large Language Models (MLLMs) (OpenAI, 2025). While this advancement fuels mass creativity, it also presents a formidable challenge to information integrity, as malicious actors can now produce highly plausible forgeries with unprecedented ease.

In response, the research community has developed a number of Image Manipulation Detection & Localization (IMDL) models (Sun et al., 2024; Wang et al., 2025; Huang et al., 2024), reaching ever-growing performances under established evaluation protocols. However, we argue that the perceived progress in this field might be a "benchmark illusion". Most existing detectors are evaluated on datasets with a narrow scope of manipulation methods (Sun et al., 2024; Huang et al., 2024) or under simplistic "train-on-one, test-on-others" protocols (Wang et al., 2025). This creates a dangerous disconnect between high reported scores and a model's true robustness in real-world scenarios, where it must confront an unpredictable and ever-expanding universe of forgery techniques. For example, an IMDL model (Wang et al., 2025) can achieve excellent performance when evaluated on a narrow set of familiar manipulation types Wang et al. (2025), creating a misleading impression of its real-world robustness. This apparent success, however, is brittle. As demonstrated in our cross-type evaluation (Table 4.2), the very same model can experience a dramatic performance collapse when confronted with unseen forgery techniques, thus revealing a critical fragility that could lead to the deployment of unreliable forensics systems.

On the other hand, existing benchmarks on AIGC manipulation have largely pushed forward the data scale Wang et al. (2025), whereas the manipulation diversity therein often lacks a systematic

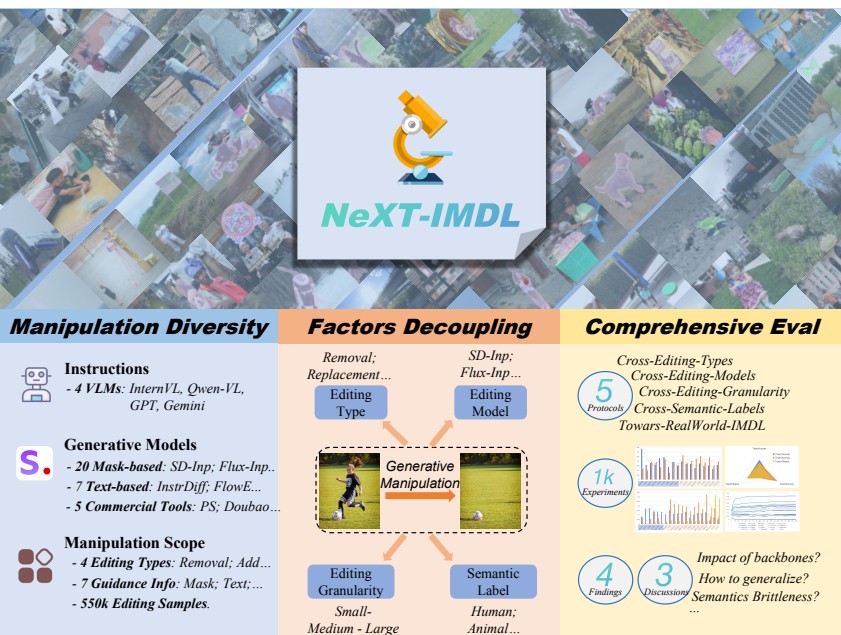

Figure 1: Overview of NeXT-IMDL. We organize predominant works in various dimensions. Moreover, we propose to decouple the AIGC IMDL task into four key aspects and build five comprehensive evaluation protocols. Insights are extracted from the results of our extensive experiments.

structure, focusing on narrow classes of manipulation models or types. The failure of state-of-the-art (SoTA) models from these benchmarks to generalize under our rigorous protocols directly exposes the "benchmark illusion". We therefore argue that the root cause of this illusion is the absence of a systematic framework to dissect the challenge and rigorously probe for model weaknesses. To address this gap, we propose the first comprehensive framework to categorize and analyze AIGC-based manipulations along four distinct and crucial axes of IMDL generalization. Each axis represents a critical potential failure point for a detector: (1) **Cross-Edit-Models** tests for overfitting to the unique "fingerprints" of specific generator architectures, a vital capability as new models are released constantly. (2) **Cross-Edit-Types** evaluates whether a model has learned the underlying concept of a forgery, or if it has merely memorized the patterns of a specific task like removal (Ekin et al., 2024; Li et al., 2025), replacement (alimama creative, 2024b; Rombach & Esser, 2022), and addition (Wasserman et al., 2024). (3) **Cross-Semantic-Labels** directly confronts the pervasive issue of "shortcut learning" (Geirhos et al., 2020), examining if a model is truly identifying manipulation artifacts or just flagging images with their unique semantic content. (4) **Cross-Edit-Granularity** assesses a model's multi-scale analysis capability, as detecting a tiny, localized edit presents a tougher challenge than spotting a large, obvious forgery. To our knowledge, this is the first work to explicitly define this multi-dimensional problem space for AIGC-based IMDL evaluation, shifting the goal from simply measuring general performances to diagnostically understanding the limits of existing approaches.

Built upon our proposed four-dimensional landscape, we introduce **NeXT-IMDL**, a large-scale diagnostic benchmark for **NeXT**-Generation **I**mage **M**anipulation **D**etection & **L**ocalization. The design of NeXT-IMDL is directly guided by the need to populate these four axes with unprecedented diversity, providing the statistical power required for meaningful cross-domain evaluation. Its composition—incorporating 32 widely-used academic and commercial editing tools, covering 4 distinct manipulation types guided by masks, text, and reference images, and spanning a wide range of semantic categories and forgery sizes—is not an end in itself, but a necessary foundation to operationalize our diagnostic approach. The benchmark is thus aimed to serve as an adversarial "stress test," revealing failure modes that remain invisible under conventional evaluation.

Our contributions are threefold: (1) Construction of a large-scale and diverse IMDL training and evaluation benchmark. NeXT-IMDL implements 32 different image editing techniques, including state-of-the-art open-source models and popular commercial tools, such as the just-released GPT-Image-1 (OpenAI, 2025) and Gemini-2.0-flash-Image (Google DeepMind Team). (2) We establish

Table 1: Comparison of previous image manipulation and detection datasets. NeXT-IMDL substantially diversifies previous datasets in terms of the number of included editing methods and manipulation diversity. Rem.

| Type | Name | Venue | Sample Number | | Traditional Manipulation | AIGC | | | |
|---|---|---|---|---|---|---|---|---|---|
| | | | Real | Manipulated | | Methods Num. | Model Type | Conditions | Manipulation |
| Traditional | Columbia(Ng & Chang, 2004) | ICME2006 | 183 | 180 | splice | — | — | — | — |
| | CASIAv2(Dong et al., 2013a) | ISIP2013 | 7491 | 5123 | copy-move;splice | — | — | — | — |
| | COVERAGE(Wen et al., 2016a) | ICIP2016 | 100 | 100 | copy-move | — | — | — | — |
| | In Wild(Huh et al., 2018) | ECCV2018 | 0 | 201 | splice | — | — | — | — |
| | NIST16(Guan et al., 2019) | WACVW2019 | 875 | 564 | copy-move;splice;remove | — | — | — | — |
| | Fantastic Reality(Kniaz et al., 2019) | NIPS2019 | 16,592 | 19,423 | copy-move;splice;remove | — | — | — | — |
| | TrainFors(Nandi et al., 2023) | ICCVW2023 | 200,000 | 800,000 | copy-move;splice;remove | — | — | — | — |
| | MIML(Qu et al., 2024) | CVPR2024 | 0 | 123,150 | splice | — | — | — | — |
| Traditional +AIGC | IMD2020 (Novozamsky et al., 2020) | WACV2020 | 35,000 | 35,000 | copy-move; splice;remove | 1 | GAN | Mask | Rep. |
| AIGC | AutoSplice(Jia et al., 2023) | CVPRW2023 | 3,621 | 2,273 | — | 1 | Diffusion | Mask | Rep. |
| | CocoGlide(Guillaro et al., 2022) | CVPR2023 | 0 | 512 | — | 1 | Diffusion | Mask | Rep. |
| | GRE(Sun et al., 2024) | MM 2024 | 0 | 228,650 | — | 5 | GAN, Diffusion | Mask | Rem., Rep., Add. |
| | GIM(Chen et al., 2025) | AAAI2025 | 1,140,000 | 1,140,000 | — | 3 | Diffusion | Mask | Rem., Rep. |
| | SID-Set(Huang et al., 2024) | CVPR2025 | 100,000 | 100,000 | — | 1 | Diffusion | Mask | Rep. |
| | OpenSDI(Wang et al., 2025) | CVPR2025 | 300,000 | 450,000 | — | 5 | Diffusion; FLUX | Mask | Rep., Rem. |
| | DiQuID(Giakoumoglou et al., 2025) | arXiv2025 | 78,000 | 95,000 | — | 6 | Diffusion; CNNs | Text, Mask | Rep., Rem. |
| **NeXT-IMDL(Ours)** | | — | **558,269** | **558,269** | — | **32** | **GAN; Diffusion; FLUX; Commercial** | **Mask, Text, Ref.Image** | **Rem., Rep., Add., Null-Text** |

i) **Types of AIGC Manipulations**: removal (*Rem.*), replacement (*Rep.*), addition (*Add.*), null-text (*Null-Text*).

5 evaluation protocols. Through our extensive experiments of SoTA models, including 6 detection & localization methods, covering representative models built for both traditional and AIGC-based manipulations, and 5 binary detection methods, we found that while achieving satisfactory results on their original settings, all 11 models fail to maintain their original performance and would drop even more significantly when conducting cross-setting evaluation. We accordingly provide 9 valuable findings to the community and aspire to inspire the development of next-generation image manipulation detection & localization models.

## 2 RELATED WORKS

**Generative image editing.** Image editing techniques have evolved significantly with generative models. Early methods like basic diffusion(Bertalmio et al., 2000; 2001; Bertalmio, 2005; Chan & Shen, 2001) and exemplar-based approaches(Criminisi et al., 2004; Jin & Ye, 2015; Kawai et al., 2015; Guo et al., 2017) have been succeeded by sophisticated GAN-based models like MAT(Li et al., 2022) and LaMa(Suvorov et al., 2022), and further by powerful diffusion models such as Stable Diffusion(Rombach et al., 2022) and ControlNet(Zhang et al., 2023b). Despite these advances, datasets for evaluating generative editing detection like DeepArt(Wang et al., 2023a), CiFAKE(Bird & Lotfi, 2024), and GenImage(Zhu et al., 2023) remain limited, as they primarily focus on image-level detection rather than the more challenging task of localization. This poses significant challenges for developing robust and generalizable detection models.

**AIGC detection datasets and benchmarks.** While AIGC detection benchmarks historically focused on facial manipulations with datasets like ForgeryNet(He et al., 2021), DeepFakeFace(Song et al., 2023), and DFFD(Cheng et al., 2024), recent works such as GenImage(Zhu et al., 2023), HiFi-IFDL(Guo et al., 2023), DiffForensics(Wang et al., 2023c), and CIFAKE(Bird & Lotfi, 2024) have expanded to general content from GANs and diffusion models(Ho et al., 2020). However, many datasets are synthetically generated (e.g., ForgeryNet(He et al., 2021), ASVspoof(Liu et al., 2023; Wang et al., 2020), DFDC(Dolhansky et al., 2020)), inadequately representing real-world deepfakes. Current limitations include outdated in-the-wild video data(Zi et al., 2020; Pu et al., 2021) and minimal linguistic diversity in audio datasets(Reimao & Tzerpos, 2019; Wang et al., 2024c; Liu et al., 2023; Pu et al., 2021), highlighting the need for more comprehensive benchmarks.

**Image manipulation detection and localization.** The paradigm for image manipulation detection primarily focuses on identifying low-level artifacts. Models like SPAN(Hu et al., 2020) and ManTra-Net(Wu et al., 2019) incorporate filters such as SRM(Zhou et al., 2018) and BayarConv(Bayar & Stamm, 2018) into VGG(Simonyan & Zisserman, 2014) backbones to extract these features, a technique also seen in MVSS-Net(Dong et al., 2021). Recent approaches tackle generalization challenges via data augmentation(Wang & Deng, 2021), adversarial training(Chen et al., 2022), and reconstruction techniques(Cao et al., 2022), while others explore frequency domain features(Jeong et al., 2022; Tan et al., 2024b) or spatial-spectral fusion(Duan et al., 2025; Wang et al., 2023b). The field has progressed to include localization, with works(Guillaro et al., 2022; Miao et al., 2023; Nguyen et al., 2024; Zhang et al., 2023a; 2024b) constructing datasets with manipulated masks. Nevertheless, many methods still struggle with overfitting(Sun et al., 2023; Zhou et al., 2023) and

Table 2: Methods included in NeXT-IMDL

| Method | ID | Rem. | Rep. | Add. | Null-Text | Venue | Condition | Data Use | Sample | Code |
|---|---|---|---|---|---|---|---|---|---|---|
| | | \multicolumn{4}{c|}{Edit-Types} | | | | | |
| Blended-Diffusion (Avrahami et al., 2022) | 1 | | ✓ | | | CVPR 2022 | Mask + Text | | 6,043 | Link |
| PbE (Yang et al., 2023) | 2 | | ✓ | | | arXiv 2022 | Mask + Ref.Image | | 8,778 | Link |
| SD2-Inpainting (AI, 2022) | 3 | ✓ | ✓ | | ✓ | HF 2022 | Mask + Src./Tar. Prompt | | 46,226 | Link |
| Inpainting-Anything (Yu et al., 2023) | 4 | ✓ | ✓ | | | arXiv 2023 | Mask + Text | | 24,129 | Link |
| anything-4.0-inpainting (Sanster, 2024) | 5 | | ✓ | | | HF 2023 | Mask + Src./Tar. Prompt | | 5,116 | Link |
| dreamshaper-8-inpainting (Lykon, 2024) | 6 | | ✓ | | ✓ | HF 2023 | Mask + Src./Tar. Prompt | | 28,945 | Link |
| SDXL-Inpainting (AI, 2023) | 7 | | ✓ | | ✓ | HF 2023 | Mask + Src./Tar. Prompt | | 26,063 | Link |
| Blended-Latent-Diffusion (Avrahami et al., 2023) | 8 | | ✓ | ✓ | | SIGGRAPH 2023 | Mask + Text | | 19,651 | Link |
| ZONE (Li et al., 2024) | 9 | ✓ | ✓ | | | CVPR 2024 | Mask + Text | | 25,725 | Link |
| PowerPaint (Zhuang et al., 2024) | 10 | ✓ | ✓ | ✓ | | ECCV 2024 | Mask + Text | Detection | 40,183 | Link |
| UltraEdit (Zhao et al., 2024a) | 11 | | ✓ | ✓ | | NeurIPS 2024 | Mask + Text | & Localization | 8,395 | Link |
| CLIPAway (Ekin et al., 2024) | 12 | ✓ | | | | NeurIPS 2024 | Mask | (Train/Val/Test) | 17,586 | Link |
| Diffree (Zhao et al., 2024b) | 13 | | ✓ | ✓ | | arXiv 2024 | Mask + Text | | 7,585 | Link |
| Kolors-Inpainting (Kwai-Kolors, 2024) | 14 | | ✓ | | | HF 2024 | Mask + Src./Tar. Prompt | | 7,258 | Link |
| SD3-Controlnet-Inpainting (alimama creative, 2024b) | 15 | | ✓ | | | HF 2024 | Mask + Src./Tar. Prompt | | 8,699 | Link |
| SD-v1.5-Inpainting (Rombach & Esser, 2022) | 16 | | ✓ | | ✓ | HF 2024 | Mask + Src./Tar. Prompt | | 29,013 | Link |
| FLUX-Inpainting (alimama creative, 2024a) | 17 | ✓ | ✓ | | ✓ | HF 2024 | Mask + Src./Tar. Prompt | | 38,180 | Link |
| HD-Painter (Manukyan et al., 2023) | 18 | | ✓ | | | ICLR 2024 | Mask + Text | | 9,257 | Link |
| RORem (Li et al., 2025) | 19 | ✓ | | | | CVPR 2025 | Mask | | 17,082 | Link |
| ACE++(Mao et al., 2025) | 20 | | ✓ | ✓ | | HF 2025 | Mask + Text | | 18,908 | Link |
| inst-inpaint (Yildirim et al., 2023) | 21 | ✓ | | | | arXiv 2023 | Text | | 19,342 | Link |
| InstructDiffusion (Geng et al., 2024) | 22 | ✓ | ✓ | | | CVPR 2024 | Text | | 28,862 | Link |
| HIVE (Zhang et al., 2024a) | 23 | ✓ | ✓ | ✓ | | arXiv 2024 | Text | Binary Detection | 43,209 | Link |
| RF-Solver-Edit (Wang et al., 2024a) | 24 | | ✓ | ✓ | | arXiv 2024 | Src./Tar. Prompt | (Train/Val/Test) | 23,836 | Link |
| FlowEdit (Kulikov et al., 2024) | 25 | | ✓ | | | arXiv 2024 | Src./Tar. Prompt | | 9,516 | Link |
| FireFlow (Deng et al., 2024) | 26 | | ✓ | ✓ | | arXiv 2024 | Src./Tar. Prompt | | 23,797 | Link |
| Paint-by-Inpaint(Wasserman et al., 2024) | 27 | | | ✓ | | CVPR 2025 | Text | | 14,385 | Link |
| Doubao Vision(ByteDance, 2023) | 28 | ✓ | | | | \ | Mask + Text | Detection | 500 | Link |
| Meitu(Meitu Network Technology Co., 2023) | 29 | ✓ | | | | \ | Mask | & Localization | 500 | Link |
| Photoshop(Inc., 2023) | 30 | ✓ | | | | \ | Mask | (Test Only) | 500 | Link |
| GPT-Image-1(OpenAI, 2025) | 31 | ✓ | ✓ | ✓ | | \ | Text | Binary Detection | 500 | Link |
| Gemini-2.0-flash(Team, 2024) | 32 | ✓ | ✓ | ✓ | | \ | Text | (Test Only) | 500 | Link |
| **Sum** | **32** | **15** | **25** | **11** | **5** | | | | **558,269** | |

i) **Venue of Methods**: HF refers to models' release date in Huggingface

generalization, especially for non-facial content, due to limited high-quality data(Dong et al., 2013a; Novozamsky et al., 2020).

# 3 NEXT-IMDL

The primary goal of the NeXT-IMDL benchmark is to provide a comprehensive and diagnostic testbed for identifying images tampered by generative models and localizing the manipulated areas. We aim to systematically operationalize the four-dimensional framework (Models, Types, Semantics, and Granularity) proposed in the introduction. This principle dictates a focus on creating structured, multi-faceted diversity, rather than merely accumulating a large volume of data. The following subsections detail our paradigm for benchmark construction, which emphasizes systematic diversity and the resulting dataset.

## 3.1 SYSTEMATIC DIVERSITY IN BENCHMARK CONSTRUCTION

Recent benchmarks such as OpenSDI (Wang et al., 2025), SIDA (Huang et al., 2024), and GRE (Sun et al., 2024) have made valuable contributions by introducing AIGC-generated data into the IMDL field. However, their construction methodologies often lack systematic and disentangled diversity. For instance, efforts (Wang et al., 2025) have focused on expanding the number of editing models, but often within the same technical family (e.g., numerous variants of Diffusion Models), which may produce shared, predictable artifacts. Other works (Wang et al., 2025; Huang et al., 2024; Sun et al., 2024) might concentrate primarily on a single guidance condition, like mask-based inpainting, thus failing to capture the full spectrum of real-world user interactions, which include text- or reference-guided edits. Consequently, the diversity within these benchmarks remains limited.

The construction of NeXT-IMDL is founded on the principle of systematic, multi-faceted diversity from the ground up. This is achieved through three core pillars: (1) Diverse Proposal Generation: We leverage a suite of four distinct VLM families to generate editing intentions, mitigating the risk of model-specific biases and ensuring a wide semantic range of manipulation proposals. (2) Diverse Manipulation Methods: We employ 32 editing models that span different architectures, training paradigms, and sources (state-of-the-art academic vs. ubiquitous commercial tools), ensuring a broad and challenging distribution of forgery traces. (3) Diverse Guidance Conditions: We simulate a variety of real-world editing scenarios by systematically incorporating multiple guidance modalities, including textual instructions, region masks, and reference images. This matrix-like approach to diversity ensures our benchmark contains structured, disentangled data, enabling the rigorous diagnostic evaluations that were previously infeasible.

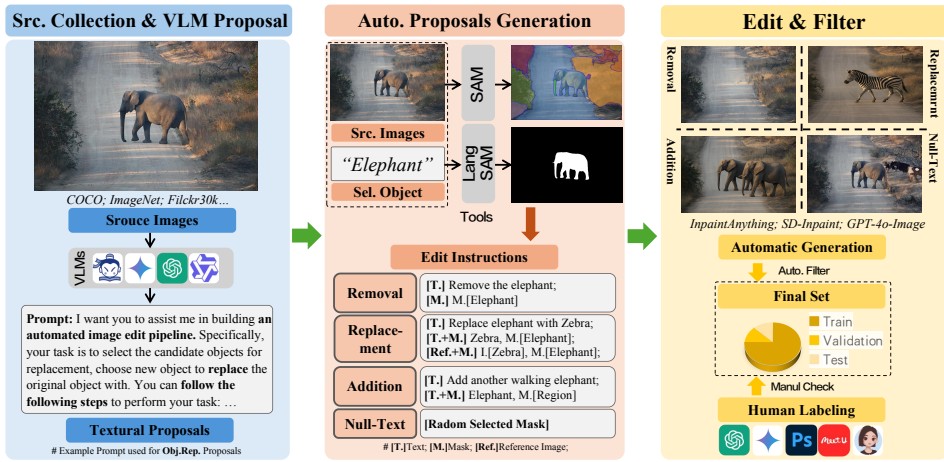

Figure 2: Generation pipeline of our proposed NeXT-IMDL dataset.

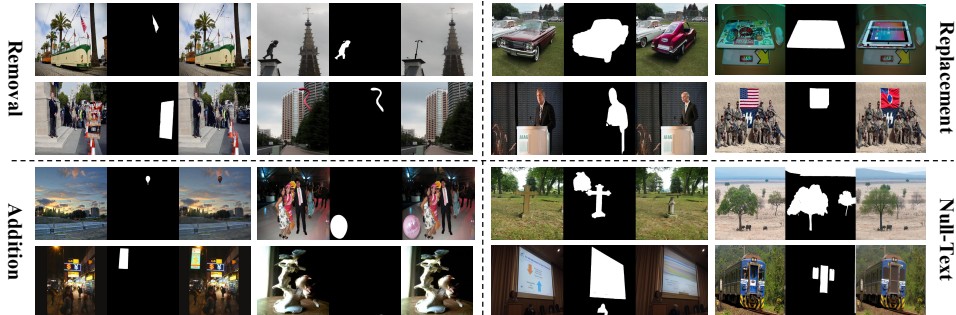

Figure 3: Samples from IMDL.

## 3.2 DATASET CONSTRUCTION

**Manipulation methods selection.** To ensure our benchmark reflects the complexity of real world, a comprehensive investigation of existing generative editing tools was conducted. Our survey spanned three key areas: latest state-of-the-art academic works, popular open-source platforms on GitHub and Huggingface Models, and widely-used commercial applications like Photoshop (Inc., 2023) and GPT-Image-1 (OpenAI, 2025). From an initial pool of over 60 candidates, we selected and deployed 32 distinct models that demonstrated high-quality results and robust instruction-following capabilities, covering a wide spectrum of underlying architectures and generative paradigms. Details of these models are summarized in Table 2.

**Original data collection and VLM proposal.** We sourced pristine images from established public datasets, including Flickr30k (Plummer et al., 2015), Microsoft COCO (Lin et al., 2014), and OpenImages V7 (Inc., 2017), to serve as the foundation for our benchmark. To generate a diverse and unbiased set of editing intentions, we then prompted a suite of four distinct VLM families (InternVL (Chen et al., 2024b;a), QWen-VL (Wang et al., 2024b; Bai et al., 2025), GPT-4o (OpenAI, 2024a;b), and Gemini (Team, 2024)) to produce manipulation instructions. The corresponding target regions for these instructions were subsequently localized using LangSAM (IndeedMiners, 2025) to create precise segmentation masks.

**Manipulation samples generation and post-filtering.** The collected source images, textual instructions, and region masks were used to generate manipulated samples across our 32 editing models, covering a wide range of guidance conditions. To guarantee the quality and realism of the final dataset, a rigorous two-stage filtering process was implemented. First, following practice in the generative image editing field (Ma et al., 2024), an automated stage discarded low-quality or failed edits using quantitative metrics like SSIM and CLIP-Similarity-Score (Radford et al., 2021). This was followed by a manual verification process to ensure all samples meet a high standard of visual fidelity. Volunteers are recruited to perform image editing using commercial tools like Photoshop (Inc., 2023). This meticulous process resulted in 558,269 high-quality manipulated samples, which were partitioned into training, validation, and testing sets with a 6:1:1 ratio.

Table 3: **Protocol-1: Cross-EM. Removal.** Image-level (detection) performance.

| | | Set - 1 | | | | Set - 2 | | | Set - 3 | | Set - 4 | Avg |
|---|---|---|---|---|---|---|---|---|---|---|---|---|
| | | Inp-Any | ZONE | CLIPAway | FLUX-Inp | SD2-Inp | PowerP | RORem | Inst-Inp | HIVE | Ins-Diff | |
| | FreqNet | 0.543 | **0.627** | 0.549 | 0.608 | 0.635 | 0.669 | 0.635 | **0.657** | 0.617 | 0.635 | 0.618 |
| | UniFD | **0.643** | 0.597 | **0.727** | **0.666** | 0.704 | 0.741 | **0.714** | 0.657 | **0.690** | **0.714** | **0.685** |
| Set - 1 | NPR | 0.512 | 0.615 | 0.611 | 0.618 | 0.705 | 0.800 | 0.685 | 0.651 | 0.678 | 0.688 | 0.656 |
| | AIDE | 0.491 | 0.555 | 0.566 | 0.477 | 0.608 | 0.691 | 0.635 | 0.611 | 0.585 | 0.565 | 0.579 |
| | FIRE | 0.585 | 0.583 | 0.585 | 0.554 | 0.630 | 0.743 | 0.676 | 0.617 | 0.626 | 0.642 | 0.624 |
| | FreqNet | 0.543 | **0.627** | 0.549 | 0.608 | 0.635 | 0.669 | 0.635 | **0.657** | 0.617 | 0.635 | 0.618 |
| | UniFD | **0.643** | 0.597 | **0.727** | **0.666** | 0.704 | 0.741 | **0.714** | 0.657 | **0.690** | **0.714** | **0.685** |
| Set - 2 | NPR | 0.512 | 0.615 | 0.611 | 0.618 | **0.705** | **0.800** | 0.685 | 0.651 | 0.678 | 0.688 | 0.656 |
| | AIDE | 0.491 | 0.555 | 0.566 | 0.477 | 0.608 | 0.691 | 0.635 | 0.611 | 0.585 | 0.565 | 0.579 |
| | FIRE | 0.585 | 0.583 | 0.585 | 0.554 | 0.630 | 0.743 | 0.676 | 0.617 | 0.626 | 0.642 | 0.624 |
| | FreqNet | 0.506 | **0.899** | 0.504 | 0.504 | 0.504 | 0.507 | 0.515 | **0.965** | **0.971** | **0.977** | 0.685 |
| | UniFD | **0.628** | 0.574 | **0.738** | **0.692** | **0.725** | 0.726 | **0.695** | 0.672 | 0.735 | 0.729 | **0.691** |
| Set - 3 | NPR | 0.497 | 0.852 | 0.526 | 0.549 | 0.537 | 0.458 | 0.540 | 0.881 | 0.873 | 0.851 | 0.656 |
| | AIDE | 0.594 | 0.693 | 0.458 | 0.455 | 0.478 | **0.746** | 0.528 | 0.839 | 0.744 | 0.672 | 0.621 |
| | FIRE | 0.509 | 0.880 | 0.495 | 0.498 | 0.500 | 0.492 | 0.511 | 0.953 | 0.962 | 0.962 | 0.676 |
| | FreqNet | 0.525 | **0.871** | 0.531 | 0.528 | 0.516 | 0.495 | 0.549 | 0.847 | 0.893 | 0.929 | **0.668** |
| | UniFD | **0.640** | 0.544 | **0.712** | **0.663** | **0.682** | **0.701** | **0.674** | 0.600 | 0.661 | 0.694 | 0.657 |
| Set - 4 | NPR | 0.513 | 0.842 | 0.645 | 0.582 | 0.559 | 0.473 | 0.606 | 0.801 | 0.767 | 0.887 | 0.668 |
| | AIDE | 0.550 | 0.603 | 0.521 | 0.501 | 0.491 | 0.590 | 0.545 | 0.603 | 0.590 | 0.809 | 0.580 |
| | FIRE | 0.506 | 0.830 | 0.506 | 0.501 | 0.500 | 0.496 | 0.508 | **0.872** | **0.895** | **0.957** | 0.657 |

Table 4: **Protocol-1: Cross-EM. Removal.** Pixel-level (localization) performance.

| | | Set-1 | | | | | | | | Set-2 | | | | | | Avg | |
|---|---|---|---|---|---|---|---|---|---|---|---|---|---|---|---|---|---|
| | | InpAny | | ZONE | | CLIPW | | FLUX-Inp | | SD2-Inp | | PowerP | | RORem | | | |
| | | IoU | F1 | IoU | F1 | IoU | F1 | IoU | F1 | IoU | F1 | IoU | F1 | IoU | F1 | IoU | F1 |
| | MVSS-Net | 0.190 | 0.223 | 0.205 | 0.245 | 0.148 | 0.185 | 0.151 | 0.188 | 0.046 | 0.061 | 0.099 | 0.126 | 0.042 | 0.057 | 0.126 | 0.155 |
| | PSCC-Net | 0.178 | 0.225 | 0.127 | 0.178 | 0.129 | 0.176 | 0.121 | 0.168 | 0.043 | 0.061 | 0.019 | 0.043 | 0.031 | 0.030 | 0.092 | 0.126 |
| Set-1 | TruFor | 0.242 | 0.286 | 0.191 | 0.235 | 0.164 | 0.202 | 0.159 | 0.195 | 0.039 | 0.051 | 0.101 | 0.128 | 0.048 | 0.066 | 0.135 | 0.166 |
| | IML-ViT | 0.236 | 0.276 | 0.156 | 0.194 | 0.165 | 0.203 | 0.192 | 0.233 | 0.038 | 0.050 | 0.037 | 0.048 | 0.016 | 0.023 | 0.120 | 0.147 |
| | Mesorch | 0.260 | 0.294 | 0.255 | 0.295 | 0.176 | 0.210 | 0.188 | 0.220 | 0.028 | 0.035 | 0.079 | 0.096 | 0.030 | 0.039 | 0.145 | 0.170 |
| | MaskCLIP | **0.428** | **0.505** | **0.398** | **0.496** | **0.372** | **0.452** | **0.359** | **0.433** | **0.091** | **0.118** | **0.260** | **0.327** | **0.108** | **0.145** | **0.288** | **0.354** |
| | MVSS-Net | 0.034 | 0.047 | 0.057 | 0.075 | 0.107 | 0.138 | 0.040 | 0.056 | 0.072 | 0.098 | 0.238 | 0.286 | 0.153 | 0.186 | 0.100 | 0.127 |
| | PSCC-Net | 0.023 | 0.034 | 0.024 | 0.036 | 0.100 | 0.141 | 0.017 | 0.025 | 0.086 | 0.124 | 0.171 | 0.227 | 0.119 | 0.163 | 0.077 | 0.107 |
| Set-2 | TruFor | 0.023 | 0.032 | 0.034 | 0.046 | 0.114 | 0.144 | 0.027 | 0.035 | 0.072 | 0.094 | 0.234 | 0.279 | 0.149 | 0.184 | 0.093 | 0.116 |
| | IML-ViT | 0.024 | 0.031 | 0.028 | 0.036 | 0.198 | 0.234 | **0.075** | **0.097** | 0.096 | 0.119 | 0.295 | 0.342 | 0.024 | 0.032 | 0.106 | 0.127 |
| | Mesorch | 0.031 | 0.041 | 0.036 | 0.045 | 0.184 | 0.218 | 0.070 | 0.088 | 0.114 | 0.139 | 0.251 | 0.291 | 0.013 | 0.017 | 0.100 | 0.120 |
| | MaskCLIP | **0.136** | **0.178** | **0.076** | **0.102** | **0.272** | **0.335** | 0.056 | 0.071 | **0.167** | **0.212** | **0.499** | **0.604** | **0.334** | **0.408** | **0.220** | **0.273** |

## 4 EXPERIMENTS AND ANALYSIS

### 4.1 EXPERIMENTS SETUP

All IMDL models  (Dong et al., 2021; Liu et al., 2021; Guillaro et al., 2022; Ma et al., 2023; Zhu et al., 2024; Wang et al., 2025) in our research scope are evaluated using the implementation in the IMDL-BenCo model zoo with default settings. We use the CAT-Net (Kwon et al., 2022) and TruFor (Guillaro et al., 2022) data protocol in IMDL-BenCo for training set organization to achieve a balance between computation resources and sample representatives. For the binary AIGC detection models studies in our research, we use their default pre-processing, training, and evaluation configurations, and rewrite their dataloaders to align with the IMDL-BenCo data protocol for a fair comparison. All training configurations and codes will be publicly available for easy reproduction.

### 4.2 EVALUATIONS, FINDINGS, AND ANALYSIS

To systematically diagnose the generalization capabilities of current IMDL models, we conduct extensive experiments under five rigorous evaluation protocols designed to probe for specific weaknesses. Each protocol creates a challenging cross-domain scenario by ensuring the training and testing sets are disjoint along a key dimension. Specifically, **Protocol 1: Cross-Edit-Models (Cross-EM)** assesses generalization to unseen manipulation tools by training on a subset of editing models ($\mathcal{M}_{train}$) and testing on a disjoint set ($\mathcal{M}_{test}$, where $\mathcal{M}_{train} \cap \mathcal{M}_{test} = \emptyset$). Similarly, **Protocol 2: Cross-Edit-Types (Cross-ET)** tests for a conceptual understanding of forgery by training on one manipulation type ($\mathcal{T}_{train}$) and evaluating on others ($\mathcal{T}_{test}$, with $\mathcal{T}_{train} \cap \mathcal{T}_{test} = \emptyset$). To probe for semantic shortcut learning, **Protocol 3: Cross-Semantic-Labels (Cross-SL)** trains on specific object categories ($\mathcal{S}_{train}$) and tests on unseen ones ($\mathcal{S}_{test}$). **Protocol 4: Cross-Edit-Granularity (Cross-EG)** challenges a model's multi-scale analysis by training and testing on disjoint sets of forgery sizes ($\mathcal{G}_{train}$ and $\mathcal{G}_{test}$). Finally, **Protocol 5: Toward-Realworld-IMDL (RealWorld-IMDL)** measures the critical "lab-to-wild" generalization gap by training on academic models ($\mathcal{M}_{academic}$) and evaluating exclusively on forgeries from commercial tools ($\mathcal{M}_{commercial}$). From the extensive results of these evaluations (as shown in

Table 5: **Protocol-2: Cross-ET**. Pixel-level (localization) performance on Protocol-2.

| | | Rem. | | | | | | Rep. | | | | | | | | | | | Add. | | | Avg |
|---|---|---|---|---|---|---|---|---|---|---|---|---|---|---|---|---|---|---|---|---|---|---|
| | | SD2-Inp | InpAny | ZONE | CLIPW | FLUX-Inp | RORem | BlenDif | PbE | Any4-Inp | Desk-Inp | SDXL-Inp | UltraE | Difffree | Ko-Inp | SD3-Contr | SD15-Inp | HDPaint | BlenLatDif | PoweaP | ACE++ | IoU |
| | | IoU | IoU | IoU | IoU | IoU | IoU | IoU | IoU | IoU | IoU | IoU | IoU | IoU | IoU | IoU | IoU | IoU | IoU | IoU | IoU | IoU |
| Rem. | MVSS-Net | 0.082 | 0.169 | 0.175 | 0.140 | 0.134 | 0.130 | 0.107 | 0.154 | 0.112 | 0.115 | 0.115 | 0.134 | 0.073 | 0.099 | 0.121 | 0.114 | 0.129 | 0.040 | 0.072 | 0.054 | 0.113 |
| | PSCC-Net | 0.044 | 0.081 | 0.061 | 0.057 | 0.061 | 0.056 | 0.066 | 0.056 | 0.063 | 0.057 | 0.068 | 0.055 | 0.026 | 0.060 | 0.053 | 0.049 | 0.065 | 0.006 | 0.009 | 0.006 | 0.050 |
| | TruFor | 0.066 | 0.199 | 0.137 | 0.156 | 0.129 | 0.138 | 0.169 | 0.173 | 0.127 | 0.118 | 0.100 | 0.127 | 0.059 | 0.087 | 0.119 | 0.110 | 0.158 | 0.063 | 0.115 | 0.054 | 0.120 |
| | IML-ViT | 0.085 | 0.267 | 0.192 | 0.193 | 0.218 | 0.209 | 0.091 | 0.246 | 0.119 | 0.120 | 0.121 | 0.126 | 0.041 | 0.043 | 0.085 | 0.110 | 0.152 | 0.087 | 0.107 | 0.038 | 0.133 |
| | Mesorch | 0.000 | 0.079 | 0.004 | 0.009 | 0.000 | 0.000 | 0.001 | 0.001 | 0.002 | 0.001 | 0.001 | 0.001 | 0.001 | 0.001 | 0.000 | 0.002 | 0.000 | 0.000 | 0.000 | 0.000 | 0.005 |
| | MaskCLIP | 0.163 | 0.410 | 0.388 | 0.375 | 0.331 | 0.306 | 0.448 | 0.385 | 0.254 | 0.256 | 0.207 | 0.235 | 0.163 | 0.152 | 0.279 | 0.230 | 0.321 | 0.193 | 0.348 | 0.123 | 0.278 |
| Rep. | MVSS-Net | 0.074 | 0.002 | 0.056 | 0.135 | 0.069 | 0.004 | 0.024 | 0.273 | 0.219 | 0.215 | 0.200 | 0.190 | 0.142 | 0.190 | 0.236 | 0.163 | 0.328 | 0.081 | 0.174 | 0.078 | 0.143 |
| | PSCC-Net | 0.093 | 0.002 | 0.075 | 0.112 | 0.045 | 0.012 | 0.329 | 0.186 | 0.164 | 0.163 | 0.184 | 0.117 | 0.107 | 0.171 | 0.227 | 0.139 | 0.166 | 0.048 | 0.112 | 0.045 | 0.125 |
| | TruFor | 0.047 | 0.000 | 0.029 | 0.123 | 0.043 | 0.001 | 0.352 | 0.220 | 0.187 | 0.171 | 0.173 | 0.217 | 0.059 | 0.155 | 0.163 | 0.126 | 0.198 | 0.069 | 0.163 | 0.059 | 0.128 |
| | IML-ViT | 0.087 | 0.003 | 0.060 | 0.140 | 0.098 | 0.007 | 0.374 | 0.262 | 0.203 | 0.204 | 0.213 | 0.241 | 0.111 | 0.210 | 0.242 | 0.161 | 0.202 | 0.077 | 0.170 | 0.096 | 0.158 |
| | Mesorch | 0.059 | 0.001 | 0.039 | 0.121 | 0.067 | 0.004 | 0.330 | 0.208 | 0.174 | 0.169 | 0.172 | 0.218 | 0.100 | 0.165 | 0.209 | 0.130 | 0.176 | 0.067 | 0.116 | 0.067 | 0.130 |
| | MaskCLIP | 0.182 | 0.030 | 0.153 | 0.344 | 0.157 | 0.021 | 0.752 | 0.603 | 0.520 | 0.477 | 0.459 | 0.582 | 0.457 | 0.470 | 0.553 | 0.361 | 0.473 | 0.273 | 0.543 | 0.278 | 0.385 |
| Add. | MVSS-Net | 0.009 | 0.001 | 0.007 | 0.018 | 0.020 | 0.001 | 0.008 | 0.041 | 0.029 | 0.033 | 0.022 | 0.036 | 0.035 | 0.029 | 0.036 | 0.018 | 0.034 | 0.181 | 0.318 | 0.280 | 0.058 |
| | PSCC-Net | 0.022 | 0.001 | 0.000 | 0.009 | 0.039 | 0.025 | 0.016 | 0.075 | 0.050 | 0.055 | 0.036 | 0.045 | 0.053 | 0.047 | 0.050 | 0.036 | 0.062 | 0.167 | 0.266 | 0.250 | 0.065 |
| | TruFor | 0.007 | 0.000 | 0.000 | 0.005 | 0.019 | 0.013 | 0.006 | 0.048 | 0.036 | 0.038 | 0.021 | 0.040 | 0.025 | 0.032 | 0.036 | 0.018 | 0.045 | 0.166 | 0.344 | 0.290 | 0.059 |
| | IML-ViT | 0.004 | 0.000 | 0.007 | 0.007 | 0.003 | 0.001 | 0.002 | 0.038 | 0.014 | 0.024 | 0.005 | 0.011 | 0.015 | 0.007 | 0.017 | 0.008 | 0.021 | 0.240 | 0.383 | 0.082 | 0.044 |
| | Mesorch | 0.003 | 0.000 | 0.002 | 0.009 | 0.008 | 0.001 | 0.004 | 0.022 | 0.014 | 0.018 | 0.009 | 0.014 | 0.019 | 0.014 | 0.013 | 0.010 | 0.020 | 0.223 | 0.353 | 0.304 | 0.053 |
| | MaskCLIP | 0.021 | 0.003 | 0.010 | 0.054 | 0.037 | 0.002 | 0.027 | 0.118 | 0.083 | 0.079 | 0.041 | 0.084 | 0.086 | 0.055 | 0.072 | 0.034 | 0.096 | 0.569 | 0.800 | 0.702 | 0.149 |

Table 6: **Protocol-3: Cross-SL.** Pixel-level (localization) performance on Protocol-3 within the replacement editing.

| Training | Model | Human | | Animal | | Object | | Avg | |
|---|---|---|---|---|---|---|---|---|---|
| | | IoU | F1 | IoU | F1 | IoU | F1 | IoU | F1 |
| Human | MVSS-Net | 0.114 | 0.168 | 0.115 | 0.169 | 0.121 | 0.171 | 0.117 | 0.169 |
| | PSCC-Net | 0.118 | 0.169 | 0.102 | 0.141 | 0.064 | 0.088 | 0.094 | 0.133 |
| | TruFor | 0.127 | 0.171 | 0.066 | 0.092 | 0.043 | 0.059 | 0.079 | 0.108 |
| | IML-ViT | 0.153 | 0.200 | 0.102 | 0.133 | 0.057 | 0.074 | 0.104 | 0.136 |
| | Mesorch | 0.128 | 0.169 | 0.054 | 0.074 | 0.032 | 0.043 | 0.071 | 0.095 |
| | MaskCLIP | 0.391 | 0.496 | 0.122 | 0.159 | 0.114 | 0.145 | 0.209 | 0.267 |
| Animal | MVSS-Net | 0.088 | 0.130 | 0.183 | 0.240 | 0.094 | 0.131 | 0.122 | 0.167 |
| | PSCC-Net | 0.093 | 0.132 | 0.164 | 0.219 | 0.079 | 0.107 | 0.112 | 0.153 |
| | TruFor | 0.039 | 0.054 | 0.168 | 0.211 | 0.036 | 0.047 | 0.081 | 0.104 |
| | IML-ViT | 0.064 | 0.088 | 0.184 | 0.228 | 0.048 | 0.062 | 0.099 | 0.126 |
| | Mesorch | 0.028 | 0.038 | 0.199 | 0.242 | 0.046 | 0.046 | 0.088 | 0.109 |
| | MaskCLIP | 0.184 | 0.242 | 0.471 | 0.570 | 0.165 | 0.206 | 0.274 | 0.339 |
| Object | MVSS-Net | 0.085 | 0.126 | 0.123 | 0.172 | 0.133 | 0.178 | 0.114 | 0.159 |
| | PSCC-Net | 0.105 | 0.150 | 0.124 | 0.167 | 0.132 | 0.175 | 0.120 | 0.164 |
| | TruFor | 0.022 | 0.031 | 0.055 | 0.072 | 0.101 | 0.128 | 0.059 | 0.077 |
| | IML-ViT | 0.063 | 0.085 | 0.080 | 0.103 | 0.108 | 0.135 | 0.083 | 0.108 |
| | Mesorch | 0.046 | 0.060 | 0.053 | 0.069 | 0.107 | 0.131 | 0.069 | 0.087 |
| | MaskCLIP | 0.224 | 0.293 | 0.252 | 0.316 | 0.361 | 0.433 | 0.279 | 0.347 |

Table 7: **Protocol-4: Cross-EG**. Pixel-level (localization) performance on Protocol-4, using all editing samples for the evaluation setup.

| Training | Model | Area - 1 | | Area - 2 | | Area - 3 | | Avg | |
|---|---|---|---|---|---|---|---|---|---|
| | | IoU | F1 | IoU | F1 | IoU | F1 | IoU | F1 |
| Area - 1 | MVSS-Net | 0.072 | 0.099 | 0.049 | 0.068 | 0.013 | 0.022 | 0.045 | 0.063 |
| | PSCC - Net | 0.083 | 0.117 | 0.081 | 0.113 | 0.031 | 0.051 | 0.065 | 0.094 |
| | TruFor | 0.052 | 0.067 | 0.020 | 0.029 | 0.002 | 0.003 | 0.025 | 0.033 |
| | IML - ViT | 0.121 | 0.148 | 0.056 | 0.074 | 0.006 | 0.010 | 0.061 | 0.077 |
| | Mesorch | 0.072 | 0.092 | 0.025 | 0.035 | 0.002 | 0.003 | 0.033 | 0.044 |
| | MaskCLIP | 0.279 | 0.344 | 0.144 | 0.195 | 0.012 | 0.022 | 0.145 | 0.187 |
| Area - 2 | MVSS-Net | 0.036 | 0.058 | 0.129 | 0.176 | 0.116 | 0.158 | 0.093 | 0.131 |
| | PSCC - Net | 0.038 | 0.062 | 0.123 | 0.171 | 0.136 | 0.186 | 0.099 | 0.140 |
| | TruFor | 0.061 | 0.080 | 0.140 | 0.171 | 0.068 | 0.093 | 0.089 | 0.115 |
| | IML - ViT | 0.105 | 0.134 | 0.218 | 0.259 | 0.121 | 0.158 | 0.148 | 0.184 |
| | Mesorch | 0.079 | 0.099 | 0.203 | 0.240 | 0.096 | 0.128 | 0.126 | 0.156 |
| | MaskCLIP | 0.208 | 0.268 | 0.406 | 0.493 | 0.198 | 0.273 | 0.271 | 0.344 |
| Area - 3 | MVSS-Net | 0.015 | 0.027 | 0.087 | 0.131 | 0.269 | 0.322 | 0.123 | 0.160 |
| | PSCC - Net | 0.016 | 0.027 | 0.076 | 0.117 | 0.247 | 0.307 | 0.113 | 0.150 |
| | TruFor | 0.030 | 0.046 | 0.124 | 0.163 | 0.286 | 0.334 | 0.147 | 0.181 |
| | IML - ViT | 0.044 | 0.062 | 0.173 | 0.212 | 0.341 | 0.381 | 0.186 | 0.218 |
| | Mesorch | 0.023 | 0.035 | 0.151 | 0.190 | 0.346 | 0.385 | 0.173 | 0.203 |
| | MaskCLIP | 0.072 | 0.105 | 0.305 | 0.384 | 0.633 | 0.719 | 0.337 | 0.403 |

Table 3.2- 7, **Bold** & Underline: best & second best results), we derive and analyze the following critical findings.

**Findings-1: IMDL models show systematic and asymmetric semantics and granularity brittleness.** As demonstrated in our Protocol-3 (in Table 6) and Protocol-4 (in Table 7) test results, models trained on one semantic or granularity set show a degradation of performance when there is a shift in either factor. For some models, this brittleness is also asymmetric. For example, the SoTA method, MaskCLIP, shows prominent generalization degradation when trained on "Human" than on others. And in Protocol-4, nearly all tested models achieve the best overall performance when trained on "Area-3". These experimental results show that the feature spaces of existing IMDL models are still prone to relying on semantic and granularity priors, showing brittleness to the bias of training data. Searching for and building unbiased extractors to leverage "semantic-agnostic" and "granularity-agnostic" features are important direction for constructing robust next-generation image manipulation detectors.

**Finding-2: The "universal donor" effect of removal forgeries.** We find that models trained exclusively on low-semantic tasks, specifically Object Removal, exhibit significantly better zero-shot performance on high-semantic tasks like Object Replacement and Addition than the reverse (Table 4.2, Fig 4). This strong asymmetry suggests that the Removal task acts as a "universal donor" for learning generalizable forgery features. By stripping away the strong semantic cue of a newly introduced object, the Removal task forces a "semantic decoupling," compelling the model to learn the fundamental, intrinsic artifacts of the generative filling process itself. This insight offers an actionable strategy for building more robust IMDL models: prioritizing low-semantic tasks in pre-training or data augmentation can instill a foundational understanding of manipulation physics over superficial semantic cues.

**Finding-3: The architectural advantage of utilizing foundation models.** During our experiments, we found that the MaskCLIP model shows a consistent and significant performance advantage over other competitors. We attribute this dominance not to incremental improvements but to its foundational Synergizing Pretrained Models (SPM) framework. Unlike monolithic models, MaskCLIP synergizes two foundation models with complementary strengths: CLIP (Radford et al., 2021), with its vast pre-training, provides robust global understanding; and MAE (He et al., 2022), which excels at learning fine-grained pixel-level representations essential for precise localization. The success

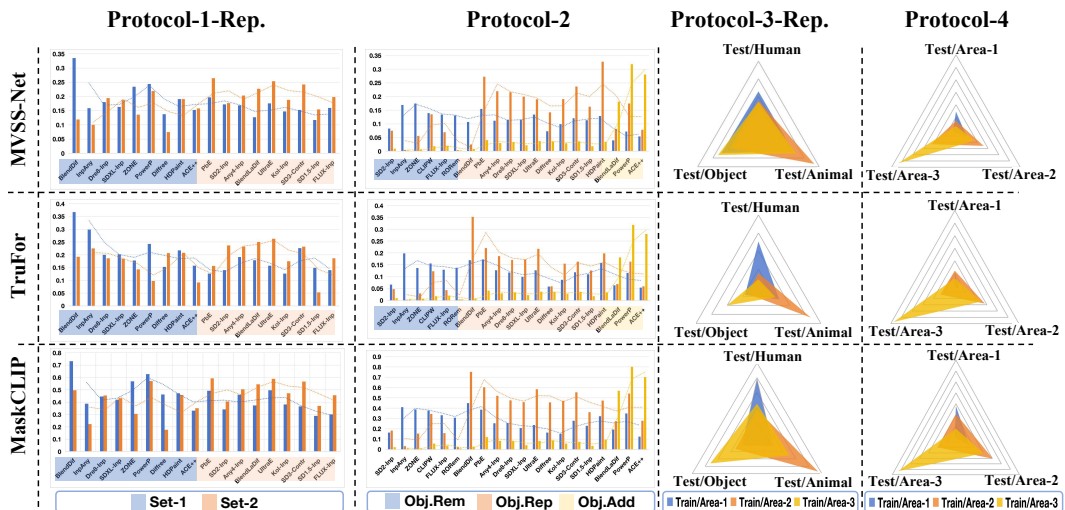

Figure 4: A brief visualization of models' IoU scores in different protocols. It can be observed from the chart that models' performances would decline to varying degrees in all four cross-scene settings.

of MaskCLIP on our diagnostic benchmark indicates a potential paradigm shift for the IMDL field, moving from designing specialized artifact extractors towards developing sophisticated methods to align and fuse the powerful, pre-existing knowledge of multiple foundation models.

**Finding-4: The "signal drowning" effect: why global AIGC detectors systematically fail at local manipulation detection.** The image-level evaluation in Protocol-1 (in Table 3.2) reveals that state-of-the-art AIGC binary classifiers (e.g., UniFD, RINE) exhibit a systemic failure, with performance often degrading to near-random chance. We identify the cause as the "signal drowning" effect. Global detectors are trained to identify subtle, holistic artifacts distributed across an entire image. In a locally manipulated image, the vast majority of the image is authentic, and its signal effectively drowns out the weak forgery signal from the small manipulated patch. Lacking a localization mechanism, the global detector cannot isolate the signal source, leading to a compromised decision. This finding highlights a fundamental distinction between the tasks of global AIGC detection and local manipulation detection (IMDL), proving that the direct application of global detectors to the IMDL problem is a flawed approach and underscoring the necessity of specialized benchmarks like NeXT-IMDL.

### 4.3 Discussions

**Discussion-1: Why is constructing a diverse dataset and benchmark critical for building the next generation IMDL models?** As indicated in previous studies in deepfake detection (Yan et al., 2024b), AIGC detection (Park & Owens, 2024), and the preceding attempt in generative IMDL (Wang et al., 2025), the performance of detectors shifts between samples generated by different models. However, for the IMDL task, performance is affected by more factors, including the semantic labels of the target object, the size of the tampered area, and so on. NeXT-IMDL, which greatly diversifies the predominant works in various dimensions and evaluation protocols, is proposed to provide a testbed for building the next-generation IMDL models. The decline of existing IMDL models when conducting cross-scene evaluation in our proposed protocols further indicates the significance of building our benchmark.

**Discussion-2: What are the characteristics of the IMDL task in the AIGC era?** Compared with traditional manipulation operations, such as splicing and copy-move (Wang et al., 2022; Ma et al., 2023), generative manipulation methods show the following features: (1) Harder to spot. Recently released generative image editing methods (AI, 2023; Rombach & Esser, 2022; Mao et al., 2025) can produce realistic samples, in which the manipulated areas are consistent with other parts of the image, and there are also few forgeries in the boundary of the real and fake areas. (2) Richer diversity. The advancement in AIGC (OpenAI, 2025) has greatly enriched the possibility of public art creation, making diverse image editing easy to reach, such as style transfer (Yu et al., 2024), and background tampering (Yu et al., 2024). Such diversity results in a large scope of semantics and forgery types,

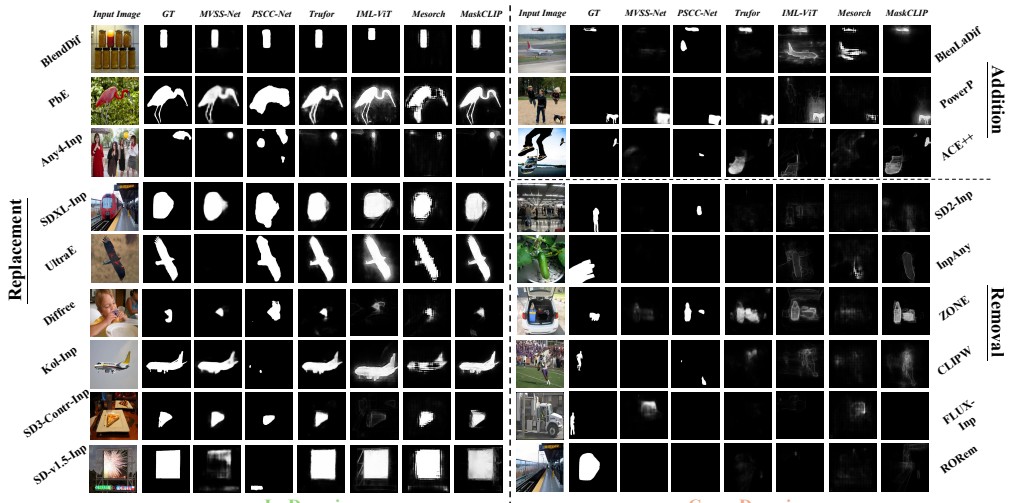

Figure 5: Qualitative results on NeXT-IMDL, Protocol-1.

making a generalizable IMDL model of urgent need. (3) Quicker evolution. The rapid development of generative models (Tian et al., 2024) has made previous SoTA AIGC detection models (Tan et al., 2024a; Yan et al., 2024a; Chu et al., 2024) quickly outdated. It is vital to develop an IMDL model that can generalize to samples manipulated by unseen models or can adapt to new forgeries with an acceptable cost.

**Discussion-3: Can existing IMDL models, mainly designed for spotting traditional manipulation forgeries, be effectively applied to solve the newly risen generative fill area localization problem?** No. As shown in our extensive experiments (e.g., Fig. 4), existing IMDL models exhibit unsatisfactory performance in in-domain evaluation, and would drop even more when evaluated on different domains. Although the previous method (Wang et al., 2025) that was especially designed for AIGC IMDL shows outstanding and relatively generalizable performance in different protocols, it's still far behind the high scores of SoTA IMDL models (Ma et al., 2023; Zhu et al., 2024) on traditional manipulation benchmarks (Dong et al., 2013b; Wen et al., 2016b; Hsu & Chang, 2006). We believe that building a universal IMDL model in the AIGC era is still waiting for future exploration.

### 4.4 OPEN QUESTIONS FOR FUTURE RESEARCH

**Questions-1: Beyond CLIP+MAE: What is the Ultimate Foundation Model for IMDL?** The success of MaskCLIP validates the robustness of large-scale pretrained feature extractors. However, is this combination the ultimate solution? Future research could explore novel ways of leveraging powerful pretrained backbones, for instance, by combining a semantic model like CLIP with a diffusion model's U-Net (Ronneberger et al., 2015) (for diffusion artifact expertise) or a dedicated segmentation model like SAM (Kirillov et al., 2023) (for boundary precision).

**Questions-2: How Can We Quantify and Actively Mitigate Semantic Bias?** We confirmed a hierarchy of semantic brittleness. This raises deeper questions: Can we develop a formal metric to quantify a model's "semantic dependency"? Furthermore, could novel training strategies, like adversarial attacks on semantic features or contrastive losses that push object and forgery features apart, be designed to explicitly enforce semantic agnosticism?

## 5 CONCLUSION

In this work, we focus on solving the problem of generative model-based image manipulation detection and localization. We start our research by identifying the four key variants when detecting AIGC manipulations: editing model, types, granularity, and the semantics of the editing area. We then propose NeXT-IMDL, a large-scale generative-based image manipulation dataset and benchmark that substantially diversifies previous works in manipulation methods, types, and evaluation protocols. We hope our findings and discussions based on our extensive experiments can bring new insights to the construction of next-generation IMDL models.

ETHICS STATEMENT

Our work, which introduces the NeXT-IMDL benchmark, is fundamentally motivated by the goal of contributing positively to society and human well-being by advancing the capabilities of image manipulation detection. We are committed to the responsible stewardship of research and have closely followed the ICLR Code of Ethics throughout this project.

**Societal Benefit:** The primary purpose of NeXT-IMDL is to provide the research community with a robust and comprehensive tool to build and test next-generation detectors against a wide array of AI-generated manipulations. By exposing the vulnerabilities of current models, we aim to spur the development of more reliable technologies to combat the spread of visual misinformation and protect information integrity.

**Data and Privacy:** The source images for our benchmark were collected from well-established, publicly available datasets (Flickr30k, Microsoft COCO, and OpenImages V7). We have used this data in a manner consistent with their original licenses and terms of use. Our data generation pipeline was automated, and no private or sensitive personal information was targeted or used. Volunteers who assisted in manual editing and quality checks did so with informed consent, and their contributions were anonymized.

**Potential for Misuse:** We acknowledge that any research in the field of forgery detection carries a potential dual-use risk. Malicious actors could theoretically study our benchmark to understand detector weaknesses and create more sophisticated forgeries. However, we firmly believe that the benefit of openly providing a challenging benchmark for defensive research significantly outweighs this risk. The rapid evolution of generative models means that robust, public-facing evaluation tools are essential for the defense to keep pace with, and ultimately get ahead of, potential threats.

**Bias and Fairness:** We have made a concerted effort to mitigate bias by incorporating a wide diversity of manipulation models (32 total), manipulation types, semantic content, and forgery sizes. We used four different VLM families to generate editing proposals to reduce the bias from any single model. Nonetheless, we recognize that biases may still exist, inherited from the large-scale source datasets or the generative models themselves. We encourage future work to further expand the diversity of the benchmark, particularly across different cultural and demographic contexts.

Our research upholds the principles of honesty and scientific excellence by transparently documenting our methodology and findings, with the ultimate goal of fostering a more secure and trustworthy digital information ecosystem.

REPRODUCIBILITY STATEMENT

We are committed to ensuring the full reproducibility of our research. To this end, we will make our dataset, code, and experimental configurations publicly available.

**Dataset:** The complete NeXT-IMDL dataset, comprising 558,269 high-quality manipulated samples along with their corresponding pristine source images and ground-truth localization masks, will be released. The release will include detailed metadata for each sample, specifying the editing model, manipulation type, guidance condition, semantic label, and forgery granularity. A detailed description of the dataset construction methodology, including source data collection, VLM-based proposal generation, and the filtering process, can be found in Section 3 of the main paper.

**Code:** We will provide open-source access to the code used for all experiments. This includes scripts for our five evaluation protocols (Cross-EM, Cross-ET, Cross-SL, Cross-EG, and RealWorld-IMDL), data loading and processing, and model evaluation. As stated in Section 4.1, the implementations of the evaluated IMDL models are based on the public IMDL-BenCo model zoo, and we will provide the necessary configurations to replicate our training and testing results.

**Experimental Details:** All details required to reproduce our experimental results are provided in Section 4 of the paper. This includes descriptions of the model training setups, the specific splits for each evaluation protocol, and the metrics used for both image-level detection and pixel-level localization. The generative models used to create the benchmark are comprehensively listed in Table 2.

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
