# A MORE EXPERIMENT RESULTS

## A.1 PROTOCOL-1: CROSS-EDIT-MODELS EVALUATION (CROSS-EM)

Table 8: **Protocol-1: Addition**. Pixel-level (localization) performance.

| Add | | Set-1 | | | | Set-2 | | Avg | |
|---|---|---|---|---|---|---|---|---|---|
| | | BlenLatDif | | PowerP | | ACE++ | | | |
| | | IoU | F1 | IoU | F1 | IoU | F1 | IoU | F1 |
| Set-1 | MVSS-Net(Dong et al., 2021) | 0.1857 | 0.2179 | 0.3108 | 0.3605 | 0.2706 | 0.2439 | 0.2557 | 0.2741 |
| | PSCC-Net(Liu et al., 2021) | 0.1613 | 0.2095 | 0.2449 | 0.3105 | 0.1121 | 0.1393 | 0.1728 | 0.2198 |
| | TruFor(Guillaro et al., 2022) | 0.1495 | 0.1714 | 0.3122 | 0.3539 | 0.1307 | 0.1537 | 0.1975 | 0.2263 |
| | IML-ViT(Ma et al., 2023) | 0.2321 | 0.2632 | 0.3745 | 0.4069 | 0.0857 | 0.0997 | 0.2308 | 0.2566 |
| | Mesorch(Zhu et al., 2024) | 0.2228 | 0.2516 | 0.3429 | 0.3803 | 0.1851 | 0.2104 | 0.2503 | 0.2808 |
| | MaskCLIP(Dong et al., 2022) | **0.5652** | **0.6386** | **0.7901** | **0.8539** | **0.3932** | **0.4400** | **0.5828** | **0.6442** |
| Set-2 | MVSS-Net(Dong et al., 2021) | 0.1124 | 0.1674 | 0.0483 | 0.0736 | 0.1082 | 0.1635 | 0.0896 | 0.1348 |
| | PSCC-Net(Liu et al., 2021) | 0.1026 | 0.1322 | 0.2134 | 0.2761 | 0.2159 | 0.2780 | 0.1773 | 0.2288 |
| | TruFor(Guillaro et al., 2022) | 0.0945 | 0.1138 | 0.2524 | 0.3009 | 0.2628 | 0.3072 | 0.2032 | 0.2406 |
| | IML-ViT(Ma et al., 2023) | 0.1263 | 0.1440 | 0.1263 | 0.1440 | 0.3315 | 0.3647 | 0.1947 | 0.2176 |
| | Mesorch(Zhu et al., 2024) | 0.1063 | 0.1196 | 0.2672 | 0.3028 | 0.2962 | 0.3321 | 0.2232 | 0.2515 |
| | MaskCLIP(Dong et al., 2022) | **0.3437** | **0.3831** | **0.7131** | **0.7845** | **0.6893** | **0.7556** | **0.5820** | **0.6411** |

Table 9: **Protocol-1: Null-Text**. Pixel-level (localization) performance.

| Training | Model | Set-1 | | | | Set-1 | | | | | | Avg | |
|---|---|---|---|---|---|---|---|---|---|---|---|---|---|
| | | Drea8-Inp | | FLUX-Inp | | SD2-Inp | | SDXL-Inp | | SD1.5-Inp | | | |
| | | IoU | F1 | IoU | F1 | IoU | F1 | IoU | F1 | IoU | F1 | IoU | F1 |
| Set-1 | MVSS-Net(Dong et al., 2021) | 0.1365 | 0.1745 | 0.1185 | 0.1547 | 0.0242 | 0.0367 | 0.0487 | 0.0683 | 0.0282 | 0.0424 | 0.0712 | 0.0953 |
| | PSCC-Net(Liu et al., 2021) | 0.1308 | 0.1818 | 0.1114 | 0.1600 | 0.0346 | 0.0544 | 0.0131 | 0.0180 | 0.0317 | 0.0486 | 0.0643 | 0.0925 |
| | TruFor(Guillaro et al., 2022)r | 0.1457 | 0.1876 | 0.1174 | 0.1559 | 0.0172 | 0.0253 | 0.0482 | 0.0655 | 0.0203 | 0.0299 | 0.0697 | 0.0928 |
| | IML-ViT(Ma et al., 2023) | 0.1305 | 0.1633 | 0.1200 | 0.1536 | 0.0082 | 0.0124 | 0.0078 | 0.0113 | 0.0226 | 0.0302 | 0.0578 | 0.0741 |
| | Mesorch(Zhu et al., 2024) | 0.1269 | 0.1547 | 0.1050 | 0.1292 | 0.0058 | 0.0082 | 0.0065 | 0.0090 | 0.0252 | 0.0324 | 0.0538 | 0.0667 |
| | MaskCLIP(Dong et al., 2022) | **0.3192** | **0.3926** | **0.2289** | **0.2871** | 0.0328 | 0.0451 | **0.0590** | **0.0781** | 0.0395 | 0.0550 | **0.1126** | **0.2007** |
| Set-2 | MVSS-Net(Dong et al., 2021) | 0.0765 | 0.1084 | 0.0227 | **0.0336** | 0.1229 | 0.1635 | 0.0857 | 0.1156 | 0.1226 | 0.1634 | 0.0740 | 0.1169 |
| | PSCC-Net(Liu et al., 2021) | 0.0788 | 0.1182 | 0.0100 | 0.0147 | 0.1123 | 0.1609 | 0.0952 | 0.1381 | 0.1105 | 0.1591 | 0.0670 | 0.1182 |
| | TruFor(Guillaro et al., 2022) | 0.0689 | 0.0948 | 0.0105 | 0.0152 | 0.1254 | 0.1643 | 0.0897 | 0.1173 | 0.1260 | 0.1659 | 0.0682 | 0.1115 |
| | IML-ViT(Ma et al., 2023) | 0.0678 | 0.0907 | 0.0128 | 0.0182 | 0.1282 | 0.1635 | 0.1279 | 0.1643 | 0.1504 | 0.1901 | 0.0696 | 0.1254 |
| | Mesorch(Zhu et al., 2024) | 0.0513 | 0.0673 | 0.0077 | 0.0099 | 0.0801 | 0.1010 | 0.0895 | 0.1134 | 0.0584 | 0.0738 | 0.0463 | 0.0731 |
| | MaskCLIP(Dong et al., 2022) | **0.1813** | **0.2319** | **0.0235** | 0.0314 | **0.2562** | **0.3231** | **0.1798** | **0.2313** | **0.2667** | **0.3388** | **0.1537** | **0.2313** |

Table 10: **Protocol-1: Removal**. Pixel-level (localization) performance.

| Removal | | Set-1 | | | | | | | | Set-2 | | | | | | Avg | |
|---|---|---|---|---|---|---|---|---|---|---|---|---|---|---|---|---|---|
| | | InpAny | | ZONE | | CLIPW | | FLUX-Inp | | SD2-Inp | | PowerP | | RORem | | | |
| | | IoU | F1 | IoU | F1 | IoU | F1 | IoU | F1 | IoU | F1 | IoU | F1 | IoU | F1 | IoU | F1 |
| Set-1 | MVSS-Net(Dong et al., 2021) | 0.1899 | 0.2230 | 0.2054 | 0.2448 | 0.1484 | 0.1851 | 0.1515 | 0.1877 | 0.0456 | 0.0611 | 0.0990 | 0.1261 | 0.0417 | 0.0570 | 0.1259 | 0.1550 |
| | PSCC-Net(Liu et al., 2021) | 0.1778 | 0.2248 | 0.1268 | 0.1778 | 0.1294 | 0.1765 | 0.1208 | 0.1678 | 0.0425 | 0.0612 | 0.0193 | 0.0432 | 0.0307 | 0.0301 | 0.0925 | 0.1259 |
| | TruFor(Guillaro et al., 2022) | 0.2425 | 0.2860 | 0.1910 | 0.2353 | 0.1642 | 0.2017 | 0.1594 | 0.1948 | 0.0387 | 0.0510 | 0.1012 | 0.1278 | 0.0480 | 0.0657 | 0.135 | 0.1660 |
| | IML-ViT(Ma et al., 2023) | 0.2360 | 0.2760 | 0.1558 | 0.1937 | 0.1649 | 0.2025 | 0.1920 | 0.2326 | 0.0380 | 0.0503 | 0.0370 | 0.0475 | 0.0163 | 0.0233 | 0.1200 | 0.1466 |
| | Mesorch(Zhu et al., 2024) | 0.2597 | 0.2944 | 0.2553 | 0.2954 | 0.1762 | 0.2099 | 0.1880 | 0.2199 | 0.0275 | 0.0352 | 0.0790 | 0.0959 | 0.0299 | 0.0392 | 0.1451 | 0.1700 |
| | MaskCLIP(Dong et al., 2022) | **0.4281** | **0.5051** | **0.3983** | **0.4961** | **0.3720** | **0.4517** | **0.3591** | **0.4326** | 0.0909 | 0.1179 | 0.2601 | 0.3270 | 0.1076 | 0.1448 | 0.2880 | 0.3536 |
| Set-2 | MVSS-Net(Dong et al., 2021) | 0.0338 | 0.0473 | 0.0567 | 0.0752 | 0.1067 | 0.1381 | 0.0399 | 0.0560 | 0.0723 | 0.0976 | 0.2382 | 0.2862 | 0.1532 | 0.1863 | 0.1001 | 0.1267 |
| | PSCC-Net(Liu et al., 2021) | 0.0234 | 0.0343 | 0.0243 | 0.0364 | 0.0996 | 0.1406 | 0.0168 | 0.0248 | 0.0861 | 0.1242 | 0.1710 | 0.2271 | 0.1189 | 0.1628 | 0.0772 | 0.1072 |
| | TruFor (Guillaro et al., 2022) | 0.0228 | 0.0319 | 0.0343 | 0.0459 | 0.1136 | 0.1441 | 0.0267 | 0.0353 | 0.0721 | 0.0937 | 0.2339 | 0.2792 | 0.1492 | 0.1838 | 0.0932 | 0.1163 |
| | IML-ViT(Ma et al., 2023) | 0.0242 | 0.0314 | 0.0278 | 0.0360 | 0.1978 | 0.2337 | 0.0753 | 0.0968 | 0.0962 | 0.1190 | 0.2949 | 0.3419 | 0.0241 | 0.0323 | 0.1058 | 0.1273 |
| | Mesorch (Zhu et al., 2024) | 0.0313 | 0.0414 | 0.0360 | 0.0455 | 0.1840 | 0.2176 | 0.0701 | 0.0885 | 0.1139 | 0.1393 | 0.2506 | 0.2906 | 0.0133 | 0.0172 | 0.0999 | 0.1200 |
| | MaskCLIP(Dong et al., 2022) | **0.1357** | **0.1782** | **0.0763** | **0.1025** | **0.2722** | **0.3354** | 0.0555 | 0.0708 | **0.1671** | **0.2123** | **0.4988** | **0.6040** | **0.3344** | **0.4079** | **0.2200** | **0.2730** |

Table 11: **Protocol-1: Add**. Image-level (detection) performance.

| Add | | Set - 1 | Set - 2 | | Set - 3 | | Set - 4 | | Avg |
|---|---|---|---|---|---|---|---|---|---|
| | | BleLatDif | PowerP | ACE++ | HIVE | FireF | RF-Sol-E | Paint-by-Inp | |
| Set - 1 | FreqNet(Cai et al., 2021) | 0.8868 | 0.8304 | 0.5655 | 0.5488 | 0.4971 | 0.4897 | 0.5555 | 0.6248 |
| | UniFD(Ojha et al., 2023) | 0.8032 | 0.7525 | 0.7041 | 0.7312 | 0.6450 | 0.6029 | 0.7102 | 0.7070 |
| | NPR(Tan et al., 2024a) | 0.9203 | 0.8708 | 0.5192 | 0.5843 | 0.4732 | 0.4672 | 0.5858 | 0.6315 |
| | AIDE(Yan et al., 2024a) | 0.7643 | 0.7639 | 0.4732 | 0.5040 | 0.4787 | 0.4738 | 0.5447 | 0.5718 |
| | FIRE(Chu et al., 2024) | 0.7872 | 0.8426 | 0.6140 | 0.5724 | 0.4864 | 0.4941 | 0.5303 | 0.6182 |
| Set - 2 | FreqNet(Cai et al., 2021) | 0.7085 | 0.6822 | 0.7120 | 0.5821 | 0.6182 | 0.6021 | 0.5836 | 0.6412 |
| | UniFD(Ojha et al., 2023) | 0.6681 | 0.6452 | 0.6454 | 0.6769 | 0.6116 | 0.5929 | 0.6236 | 0.6377 |
| | NPR(Tan et al., 2024a) | 0.6817 | 0.6435 | 0.7565 | 0.5619 | 0.6531 | 0.6361 | 0.5645 | 0.6425 |
| | AIDE(Yan et al., 2024a) | 0.3440 | 0.4756 | 0.5803 | 0.5358 | 0.5070 | 0.5251 | 0.4809 | 0.4927 |
| | FIRE(Chu et al., 2024) | 0.5224 | 0.5572 | 0.5329 | 0.5561 | 0.5951 | 0.6468 | 0.5807 | 0.5702 |
| Set - 3 | FreqNet(Cai et al., 2021) | 0.5743 | 0.5518 | 0.6284 | 0.8792 | 0.7217 | 0.5837 | 0.5404 | 0.6399 |
| | UniFD(Ojha et al., 2023) | 0.6506 | 0.6044 | 0.6349 | 0.7912 | 0.6714 | 0.7131 | 0.6345 | 0.6714 |
| | NPR(Tan et al., 2024a) | 0.6885 | 0.6507 | 0.5843 | 0.8025 | 0.8278 | 0.6036 | 0.5476 | 0.6721 |
| | AIDE(Yan et al., 2024a) | 0.4534 | 0.6566 | 0.5485 | 0.7511 | 0.8194 | 0.7139 | 0.4322 | 0.6250 |
| | FIRE(Chu et al., 2024) | 0.5029 | 0.5097 | 0.5304 | 0.8524 | 0.8021 | 0.5889 | 0.5274 | 0.6163 |
| Set - 4 | FreqNet(Cai et al., 2021) | 0.6113 | 0.5846 | 0.5731 | 0.5901 | 0.5477 | 0.5911 | 0.5844 | 0.5832 |
| | UniFD(Ojha et al., 2023) | 0.6229 | 0.5985 | 0.6151 | 0.7460 | 0.6681 | 0.7161 | 0.6200 | 0.6552 |
| | NPR(Tan et al., 2024a) | 0.5224 | 0.5299 | 0.5213 | 0.5297 | 0.5257 | 0.5302 | 0.5278 | 0.5267 |
| | AIDE(Yan et al., 2024a) | 0.4747 | 0.6061 | 0.5362 | 0.5944 | 0.7416 | 0.7518 | 0.4445 | 0.5928 |
| | FIRE(Chu et al., 2024) | 0.5219 | 0.5522 | 0.5402 | 0.5753 | 0.5918 | 0.6445 | 0.5840 | 0.5728 |

Table 12: **Protocol-1: Null-Text**. Image-level (detection) performance.

| Null-Text | | Set-1 | | Set-2 | | | Avg |
|---|---|---|---|---|---|---|---|
| | | Drea-8-Inp | FLUX-Inp | SD2-Inp | SDXL-Inp | SD1.5-Inp | |
| Set-1 | FreqNet(Cai et al., 2021) | 0.7458 | 0.6279 | 0.6314 | 0.5787 | 0.5445 | 0.6257 |
| | UniFD(Ojha et al., 2023) | 0.6596 | 0.6335 | 0.6414 | 0.5885 | 0.6554 | 0.6357 |
| | NPR(Tan et al., 2024a) | 0.7712 | 0.6525 | 0.6623 | 0.5624 | 0.5880 | 0.6473 |
| | AIDE(Yan et al., 2024a) | 0.7021 | 0.7242 | 0.6870 | 0.5808 | 0.6624 | 0.6713 |
| | FIRE(Chu et al., 2024) | 0.6699 | 0.6109 | 0.5481 | 0.5377 | 0.5263 | 0.5786 |
| Set-2 | FreqNet(Cai et al., 2021) | 0.5994 | 0.5630 | 0.6072 | 0.5985 | 0.5732 | 0.5883 |
| | UniFD(Ojha et al., 2023) | 0.6587 | 0.6497 | 0.6736 | 0.6079 | 0.6870 | 0.6554 |
| | NPR(Tan et al., 2024a) | 0.6731 | 0.5847 | 0.6831 | 0.6727 | 0.6915 | 0.6610 |
| | AIDE(Yan et al., 2024a) | 0.6691 | 0.6127 | 0.7140 | 0.6364 | 0.7480 | 0.6760 |
| | FIRE(Chu et al., 2024) | 0.5695 | 0.5367 | 0.6255 | 0.5918 | 0.6591 | 0.5965 |

Table 13: **Protocol-1: Removal**. Image-level (detection) performance.

| Removal | | Set - 1 | | | | Set - 2 | | | Set - 3 | | Set - 4 | Avg |
|---|---|---|---|---|---|---|---|---|---|---|---|---|
| | | Inp-Any | ZONE | CLIPAway | FLUX-Inp | SD2-Inp | PowerP | RORem | Inst-Inp | HIVE | Ins-Diff | |
| Set - 1 | FreqNet(Cai et al., 2021) | 0.5434 | 0.6275 | 0.5491 | 0.6082 | 0.6346 | 0.6693 | 0.6346 | 0.6573 | 0.6165 | 0.6352 | 0.6176 |
| | UniFD(Ojha et al., 2023) | 0.6427 | 0.5974 | 0.7274 | 0.6664 | 0.7042 | 0.7405 | 0.7141 | 0.6567 | 0.6903 | 0.7140 | 0.6854 |
| | NPR(Tan et al., 2024a) | 0.5121 | 0.6146 | 0.6110 | 0.6177 | 0.7050 | 0.7999 | 0.6853 | 0.6515 | 0.6776 | 0.6877 | 0.6562 |
| | AIDE(Yan et al., 2024a) | 0.4909 | 0.5551 | 0.5658 | 0.4774 | 0.6081 | 0.6908 | 0.6354 | 0.6108 | 0.5854 | 0.5654 | 0.5785 |
| | FIRE(Chu et al., 2024) | 0.5851 | 0.5826 | 0.5850 | 0.5541 | 0.6298 | 0.7432 | 0.6760 | 0.6173 | 0.6259 | 0.6419 | 0.6241 |
| Set - 2 | FreqNet(Cai et al., 2021) | 0.5434 | 0.6275 | 0.5491 | 0.6082 | 0.6346 | 0.6693 | 0.6346 | 0.6573 | 0.6165 | 0.6352 | 0.6176 |
| | UniFD(Ojha et al., 2023) | 0.6427 | 0.5974 | 0.7274 | 0.6664 | 0.7042 | 0.7405 | 0.7141 | 0.6567 | 0.6903 | 0.7140 | 0.6854 |
| | NPR(Tan et al., 2024a) | 0.5121 | 0.6146 | 0.6110 | 0.6177 | 0.7050 | 0.7999 | 0.6853 | 0.6515 | 0.6776 | 0.6877 | 0.6562 |
| | AIDE(Yan et al., 2024a) | 0.4909 | 0.5551 | 0.5658 | 0.4774 | 0.6081 | 0.6908 | 0.6354 | 0.6108 | 0.5854 | 0.5654 | 0.5785 |
| | FIRE(Chu et al., 2024) | 0.5851 | 0.5826 | 0.5850 | 0.5541 | 0.6298 | 0.7432 | 0.6760 | 0.6173 | 0.6259 | 0.6419 | 0.6241 |
| Set - 3 | FreqNet(Cai et al., 2021) | 0.5059 | 0.8992 | 0.5043 | 0.5044 | 0.5044 | 0.5072 | 0.5150 | 0.9645 | 0.9712 | 0.9767 | 0.6853 |
| | UniFD(Ojha et al., 2023) | 0.6277 | 0.5737 | 0.7383 | 0.6917 | 0.7250 | 0.7258 | 0.6952 | 0.672 | 0.7353 | 0.7288 | 0.6914 |
| | NPR(Tan et al., 2024a) | 0.4972 | 0.8522 | 0.5260 | 0.5487 | 0.5371 | 0.4580 | 0.5399 | 0.8808 | 0.8726 | 0.8506 | 0.6563 |
| | AIDE(Yan et al., 2024a) | 0.5936 | 0.6925 | 0.4581 | 0.4548 | 0.4785 | 0.7463 | 0.5279 | 0.8394 | 0.7437 | 0.6716 | 0.6206 |
| | FIRE(Chu et al., 2024) | 0.5089 | 0.8797 | 0.4953 | 0.4980 | 0.4998 | 0.4923 | 0.5115 | 0.9527 | 0.9623 | 0.9617 | 0.6762 |
| Set - 4 | FreqNet(Cai et al., 2021) | 0.5250 | 0.8706 | 0.5312 | 0.5280 | 0.5156 | 0.4946 | 0.5491 | 0.8467 | 0.8926 | 0.9287 | 0.6682 |
| | UniFD(Ojha et al., 2023) | 0.6402 | 0.5445 | 0.7115 | 0.6630 | 0.6825 | 0.7007 | 0.6740 | 0.5997 | 0.6610 | 0.6935 | 0.6571 |
| | NPR(Tan et al., 2024a) | 0.5131 | 0.8418 | 0.6455 | 0.5824 | 0.5588 | 0.4735 | 0.6062 | 0.8013 | 0.7673 | 0.8870 | 0.6677 |
| | AIDE(Yan et al., 2024a) | 0.5504 | 0.6025 | 0.5210 | 0.5007 | 0.4910 | 0.5904 | 0.5450 | 0.6033 | 0.5902 | 0.8091 | 0.5804 |
| | FIRE(Chu et al., 2024) | 0.5057 | 0.8297 | 0.5060 | 0.5012 | 0.5004 | 0.4961 | 0.5076 | 0.8721 | 0.8947 | 0.9565 | 0.6570 |

Table 14: **Protocol-1: Replacement.** Image-level (detection) performance.

| Training | Model | BlenDif | InpAny | Dreas-Inp | SDXL-Inp | ZONE | Power | Difree | HD-Paint | ACE++ | PbE | SD2-Inp | Any+-Inp | BlendLaIDif | UltraE | Kol-Inp | SD3-Cont | SD1.5-Inp | FLUX-Inp | Ins-Diff | Flowt | Fisrt | HiVE | RF-Sol-E | Avg Accuracy |
|---|---|---|---|---|---|---|---|---|---|---|---|---|---|---|---|---|---|---|---|---|---|---|---|---|---|
| Set-1 | FreqNet(Cai et al., 2021) | 0.7630 | 0.5439 | 0.7054 | 0.6336 | 0.7349 | 0.6770 | 0.7498 | 0.7157 | 0.5516 | 0.7128 | 0.6636 | 0.6628 | 0.7420 | 0.6357 | 0.6168 | 0.6756 | 0.6121 | 0.6499 | 0.6385 | 0.5554 | 0.5959 | 0.7684 | 0.5468 | 0.6647 |
| | UniFD(Ojha et al., 2023) | 0.7217 | 0.6732 | 0.7314 | 0.7120 | 0.6701 | 0.7062 | 0.6125 | 0.7214 | 0.6500 | 0.7435 | 0.7256 | 0.7570 | 0.7459 | 0.7353 | 0.6826 | 0.6818 | 0.7346 | 0.7114 | 0.7271 | 0.6358 | 0.6622 | 0.7371 | 0.6290 | 0.7003 |
| | NPR(Tan et al., 2024a) | 0.8284 | 0.8284 | 0.7430 | 0.6733 | 0.7442 | 0.7097 | 0.7057 | 0.7625 | 0.6077 | 0.7731 | 0.6856 | 0.7522 | 0.7850 | 0.6474 | 0.6533 | 0.6316 | 0.6754 | 0.6772 | 0.7873 | 0.6190 | 0.6536 | 0.7715 | 0.5784 | 0.7084 |
| | AIDE(Yan et al., 2024a) | 0.7538 | 0.5730 | 0.6289 | 0.5397 | 0.6636 | 0.7122 | 0.5744 | 0.6181 | 0.5778 | 0.5592 | 0.5950 | 0.5517 | 0.5809 | 0.5222 | 0.5306 | 0.5695 | 0.5532 | 0.5342 | 0.6498 | 0.7070 | 0.7021 | 0.6577 | 0.6052 | 0.6069 |
| | FIRE(Chu et al., 2024) | 0.6553 | 0.5756 | 0.6313 | 0.6182 | 0.7358 | 0.6707 | 0.7267 | 0.6349 | 0.5593 | 0.6318 | 0.6126 | 0.6155 | 0.6097 | 0.5767 | 0.5942 | 0.6065 | 0.5896 | 0.5726 | 0.7567 | 0.5512 | 0.5210 | 0.7320 | 0.5423 | 0.6226 |
| Set-2 | FreqNet(Cai et al., 2021) | 0.6992 | 0.6992 | 0.7163 | 0.6341 | 0.5969 | 0.6970 | 0.6091 | 0.7442 | 0.5637 | 0.7550 | 0.7014 | 0.7034 | 0.7775 | 0.6545 | 0.6378 | 0.7047 | 0.6393 | 0.6836 | 0.6385 | 0.5705 | 0.5629 | 0.6251 | 0.5261 | 0.6583 |
| | UniFD(Ojha et al., 2023) | 0.7319 | 0.6619 | 0.7584 | 0.7284 | 0.6705 | 0.7220 | 0.5757 | 0.7509 | 0.6440 | 0.7916 | 0.7511 | 0.7870 | 0.7924 | 0.7617 | 0.6834 | 0.7029 | 0.7714 | 0.7387 | 0.7505 | 0.6492 | 0.6581 | 0.7773 | 0.6190 | 0.7164 |
| | NPR(Tan et al., 2024a) | 0.7068 | 0.7068 | 0.7683 | 0.6311 | 0.6594 | 0.6963 | 0.6459 | 0.7934 | 0.5593 | 0.8520 | 0.7164 | 0.7999 | 0.8566 | 0.6951 | 0.6453 | 0.7567 | 0.7368 | 0.6781 | 0.7749 | 0.5715 | 0.5244 | 0.7141 | 0.4876 | 0.6874 |
| | AIDE(Yan et al., 2024a) | 0.6747 | 0.4440 | 0.6692 | 0.6693 | 0.4124 | 0.4511 | 0.4444 | 0.7235 | 0.4802 | 0.7902 | 0.6847 | 0.7185 | 0.7711 | 0.7203 | 0.6608 | 0.6945 | 0.7618 | 0.6874 | 0.4825 | 0.3587 | 0.4024 | 0.4347 | 0.3807 | 0.5877 |
| | FIRE(Chu et al., 2024) | 0.5848 | 0.5726 | 0.6397 | 0.6028 | 0.5885 | 0.6675 | 0.5766 | 0.6448 | 0.5540 | 0.6987 | 0.6605 | 0.6653 | 0.6959 | 0.7124 | 0.6164 | 0.6967 | 0.6543 | 0.6277 | 0.6241 | 0.5399 | 0.4876 | 0.6052 | 0.5021 | 0.6182 |
| Set-3 | FreqNet(Cai et al., 2021) | 0.5158 | 0.5401 | 0.5815 | 0.6147 | 0.7349 | 0.4870 | 0.7305 | 0.5942 | 0.5839 | 0.5781 | 0.5981 | 0.5400 | 0.6104 | 0.5695 | 0.5528 | 0.5764 | 0.5875 | 0.5995 | 0.7856 | 0.6651 | 0.7237 | 0.7636 | 0.5932 | 0.6142 |
| | UniFD(Ojha et al., 2023) | 0.5787 | 0.6229 | 0.6752 | 0.6872 | 0.6482 | 0.6657 | 0.6056 | 0.6831 | 0.6383 | 0.6965 | 0.6781 | 0.7163 | 0.6742 | 0.7045 | 0.6461 | 0.6535 | 0.6711 | 0.6388 | 0.7409 | 0.7280 | 0.6942 | 0.7866 | 0.7369 | 0.6770 |
| | NPR(Tan et al., 2024a) | 0.5255 | 0.5685 | 0.5927 | 0.6112 | 0.7060 | 0.5559 | 0.6848 | 0.6037 | 0.5819 | 0.6451 | 0.5822 | 0.5880 | 0.5930 | 0.5673 | 0.5896 | 0.5913 | 0.6193 | 0.6025 | 0.7605 | 0.7325 | 0.7485 | 0.7230 | 0.6558 | 0.6273 |
| | AIDE(Yan et al., 2024a) | 0.4632 | 0.5809 | 0.4842 | 0.4950 | 0.6934 | 0.6731 | 0.5885 | 0.4736 | 0.5246 | 0.5581 | 0.4785 | 0.4831 | 0.4954 | 0.4831 | 0.4966 | 0.5080 | 0.5014 | 0.4953 | 0.8127 | 0.8845 | 0.8488 | 0.6213 | 0.6970 | 0.5791 |
| | FIRE(Chu et al., 2024) | 0.5352 | 0.5495 | 0.5351 | 0.5422 | 0.5554 | 0.5644 | 0.5351 | 0.5474 | 0.5480 | 0.5333 | 0.5400 | 0.5279 | 0.5302 | 0.5466 | 0.5553 | 0.5353 | 0.5300 | 0.5534 | 0.8055 | 0.6180 | 0.6199 | 0.5495 | 0.6090 | 0.5629 |
| Set-4 | FreqNet(Cai et al., 2021) | 0.4704 | 0.5473 | 0.5909 | 0.5452 | 0.6696 | 0.5123 | 0.6818 | 0.5833 | 0.5641 | 0.5514 | 0.6003 | 0.6228 | 0.5674 | 0.5477 | 0.5871 | 0.5731 | 0.5639 | 0.5666 | 0.7213 | 0.5939 | 0.5591 | 0.7271 | 0.6159 | 0.5897 |
| | UniFD(Ojha et al., 2023) | 0.5368 | 0.6040 | 0.6577 | 0.6748 | 0.6291 | 0.6661 | 0.5800 | 0.6620 | 0.5859 | 0.6869 | 0.6566 | 0.7291 | 0.6608 | 0.6850 | 0.6340 | 0.6280 | 0.6407 | 0.6089 | 0.7392 | 0.6599 | 0.6498 | 0.8192 | 0.7208 | 0.6572 |
| | NPR(Tan et al., 2024a) | 0.4750 | 0.5337 | 0.5765 | 0.5253 | 0.6873 | 0.5401 | 0.6429 | 0.5857 | 0.5427 | 0.5352 | 0.5734 | 0.6433 | 0.5791 | 0.5323 | 0.5611 | 0.5215 | 0.5286 | 0.5090 | 0.7392 | 0.6245 | 0.5581 | 0.8419 | 0.7266 | 0.5906 |
| | AIDE(Yan et al., 2024a) | 0.3233 | 0.6120 | 0.4393 | 0.4419 | 0.6692 | 0.6833 | 0.6176 | 0.4037 | 0.5504 | 0.3875 | 0.4204 | 0.4260 | 0.3811 | 0.3977 | 0.4552 | 0.4480 | 0.3771 | 0.3689 | 0.6412 | 0.7579 | 0.7302 | 0.7213 | 0.7239 | 0.5207 |
| | FIRE(Chu et al., 2024) | 0.4888 | 0.5295 | 0.5555 | 0.5353 | 0.7088 | 0.5464 | 0.6946 | 0.5611 | 0.5165 | 0.5322 | 0.5431 | 0.5718 | 0.5195 | 0.5282 | 0.5448 | 0.5095 | 0.5343 | 0.5085 | 0.7811 | 0.5516 | 0.5808 | 0.7838 | 0.6327 | 0.5765 |

Table 15: **Protocol-1: Replacement.** Pixel-level (localization) **IoU** performance.

| Training | Model | BlenDif | InpAny | Dreas-Inp | SDXL-Inp | ZONE | Power | Difree | HDPaint | ACE++ | PbE | SD2-Inp | Any+-Inp | BlenLaDif | UltraE | Kol-Inp | SD3-Cont | SD1.5-Inp | FLUX-Inp | Avg IoU |
|---|---|---|---|---|---|---|---|---|---|---|---|---|---|---|---|---|---|---|---|---|
| Set-1 | MVSS-Net(Dong et al., 2021) | 0.3355 | 0.1586 | 0.1802 | 0.1629 | 0.2341 | 0.2442 | 0.1380 | 0.1908 | 0.1527 | 0.1966 | 0.1724 | 0.1689 | 0.1266 | 0.1756 | 0.1476 | 0.1522 | 0.1175 | 0.1593 | 0.1785 |
| | PSCC-Net(Liu et al., 2021) | 0.0925 | 0.2979 | 0.1785 | 0.1466 | 0.1785 | 0.1581 | 0.1529 | 0.1701 | 0.1352 | 0.1201 | 0.1154 | 0.1676 | 0.1309 | 0.1417 | 0.1262 | 0.1559 | 0.1341 | 0.1343 | 0.1499 |
| | TruForr(Guillaro et al., 2022) | 0.3674 | 0.2988 | 0.2002 | 0.2018 | 0.1783 | 0.2423 | 0.1517 | 0.2179 | 0.1577 | 0.1267 | 0.1397 | 0.1917 | 0.1792 | 0.1571 | 0.1262 | 0.2266 | 0.1489 | 0.1397 | 0.1918 |
| | IML-ViT(Ma et al., 2023) | 0.3839 | 0.3182 | 0.2122 | 0.2015 | 0.1603 | 0.2265 | 0.1632 | 0.2217 | 0.1688 | 0.1323 | 0.1532 | 0.1924 | 0.1663 | 0.1552 | 0.1893 | 0.2414 | 0.1560 | 0.1532 | 0.1998 |
| | Mesorch(Zhu et al., 2024) | 0.3590 | 0.1601 | 0.1687 | 0.1472 | 0.1922 | 0.2520 | 0.0932 | 0.1858 | 0.1178 | 0.1905 | 0.1132 | 0.1585 | 0.1220 | 0.1593 | 0.1215 | 0.1271 | 0.0953 | 0.1004 | 0.1591 |
| | MaskCLIP(Dong et al., 2022) | 0.7318 | 0.3875 | 0.4458 | 0.4203 | 0.5683 | 0.6281 | 0.4621 | 0.4716 | 0.3315 | 0.4930 | 0.3407 | 0.4608 | 0.3729 | 0.4954 | 0.3815 | 0.3645 | 0.2897 | 0.3013 | 0.4415 |
| Set-2 | MVSS-Net(Dong et al., 2021) | 0.1194 | 0.1007 | 0.1941 | 0.1875 | 0.1359 | 0.2190 | 0.0748 | 0.1908 | 0.1577 | 0.2639 | 0.1764 | 0.2032 | 0.2271 | 0.2534 | 0.1882 | 0.2419 | 0.1546 | 0.1982 | 0.1826 |
| | PSCC-Net(Liu et al., 2021) | 0.0735 | 0.0702 | 0.1552 | 0.1713 | 0.1099 | 0.1799 | 0.0589 | 0.1540 | 0.1295 | 0.1846 | 0.1595 | 0.1560 | 0.1699 | 0.2104 | 0.1772 | 0.2322 | 0.1418 | 0.1651 | 0.1500 |
| | TruForr(Guillaro et al., 2022) | 0.1921 | 0.2255 | 0.1863 | 0.1843 | 0.1434 | 0.0975 | 0.2069 | 0.2075 | 0.0920 | 0.1570 | 0.2368 | 0.2322 | 0.2505 | 0.2627 | 0.1745 | 0.2321 | 0.0530 | 0.1862 | 0.1845 |
| | IML-ViT(Ma et al., 2023) | 0.2101 | 0.1924 | 0.2024 | 0.1923 | 0.1226 | 0.0952 | 0.2019 | 0.2098 | 0.0794 | 0.1575 | 0.2676 | 0.2402 | 0.2517 | 0.2813 | 0.2130 | 0.2189 | 0.0670 | 0.1761 | 0.1877 |
| | Mesorch(Zhu et al., 2024) | 0.1611 | 0.1919 | 0.1863 | 0.1819 | 0.1542 | 0.1130 | 0.1873 | 0.1991 | 0.0907 | 0.1459 | 0.2377 | 0.2329 | 0.2419 | 0.2515 | 0.1821 | 0.1913 | 0.0579 | 0.1619 | 0.1760 |
| | MaskCLIP(Dong et al., 2022) | 0.4971 | 0.2215 | 0.4536 | 0.4321 | 0.3060 | 0.5709 | 0.1764 | 0.4579 | 0.3513 | 0.5926 | 0.4066 | 0.5038 | 0.5439 | 0.5891 | 0.4715 | 0.5658 | 0.3699 | 0.4556 | 0.4425 |

Table 16: **Protocol-1: Replacement.** Pixel-level (localization) **F1** performance.

| Training | Model | Set-1 | | | | | | | | | | Set-2 | | | | | | | | Avg |
|---|---|---|---|---|---|---|---|---|---|---|---|---|---|---|---|---|---|---|---|---|
| | | BlendDif | InpAny | Dreas-Inp | SDXL-Inp | ZONE | PowerP | Diffree | HDPaint | ACE++ | PbE | SD2-Inp | Anya4-Inp | BlendaDif | UltraE | Kol-Inp | SD3-Cont | SD1.5-Inp | FLUX-Inp | IOU |
| Set-1 | MVSS-Net (Dong et al., 2021) | 0.3676 | 0.1953 | 0.2178 | 0.2003 | 0.2697 | 0.2917 | 0.1737 | 0.2307 | 0.1905 | 0.2390 | 0.1724 | 0.2074 | 0.1617 | 0.2184 | 0.1806 | 0.1876 | 0.1484 | 0.1593 | 0.2118 |
| | PSCC-Net (Liu et al., 2021) | 0.1331 | 0.3429 | 0.2373 | 0.1946 | 0.1898 | 0.2112 | 0.2022 | 0.2231 | 0.1771 | 0.1676 | 0.1564 | 0.2176 | 0.1696 | 0.1869 | 0.1692 | 0.2070 | 0.1799 | 0.1790 | 0.1969 |
| | TruFor (Guillaro et al., 2022) | 0.3985 | 0.3438 | 0.2345 | 0.2385 | 0.2139 | 0.2748 | 0.1850 | 0.2555 | 0.1883 | 0.1532 | 0.1651 | 0.2278 | 0.2076 | 0.1905 | 0.1692 | 0.2674 | 0.1788 | 0.1651 | 0.2264 |
| | IML-ViT (Ma et al., 2023) | 0.4130 | 0.3674 | 0.2542 | 0.2419 | 0.2003 | 0.2631 | 0.2047 | 0.2665 | 0.2045 | 0.1639 | 0.1874 | 0.2323 | 0.2080 | 0.1893 | 0.2323 | 0.2892 | 0.1922 | 0.1874 | 0.2388 |
| | Mesorch (Zhu et al., 2024) | 0.3876 | 0.1925 | 0.1993 | 0.1774 | 0.2238 | 0.2939 | 0.1160 | 0.2206 | 0.1445 | 0.2293 | 0.1396 | 0.1873 | 0.1506 | 0.1940 | 0.1438 | 0.1540 | 0.1179 | 0.1204 | 0.1885 |
| | MaskCLIP (Dong et al., 2022) | **0.7976** | **0.4669** | **0.5225** | **0.4950** | **0.6383** | **0.7207** | **0.5568** | **0.5546** | **0.3936** | **0.5830** | **0.4084** | **0.5391** | **0.4519** | **0.5779** | **0.4342** | **0.3488** | **0.3578** | **0.3578** | **0.5156** |
| Set-2 | MVSS-Net(Dong et al., 2021) | 0.1432 | 0.1287 | 0.2314 | 0.2231 | 0.1603 | 0.2609 | 0.0930 | 0.2296 | 0.1917 | 0.3152 | 0.2145 | 0.2434 | 0.2700 | 0.2960 | 0.2178 | 0.2830 | 0.1902 | 0.2350 | 0.2182 |
| | PSCC-Net(Liu et al., 2021) | 0.0908 | 0.0926 | 0.2023 | 0.2144 | 0.1364 | 0.2280 | 0.0778 | 0.2015 | 0.1664 | 0.2419 | 0.2107 | 0.2057 | 0.2242 | 0.2683 | 0.2200 | 0.2890 | 0.1908 | 0.2169 | 0.1932 |
| | TruFor(Guillaro et al., 2022) | 0.2195 | 0.2627 | 0.2187 | 0.2149 | 0.1707 | 0.1136 | 0.2440 | 0.2440 | 0.1159 | 0.1887 | 0.2738 | 0.2727 | 0.2901 | 0.3080 | 0.1987 | 0.2727 | 0.0642 | 0.2196 | 0.2162 |
| | IML-ViT(Ma et al., 2023) | 0.2380 | 0.2286 | 0.2424 | 0.2312 | 0.1510 | 0.1192 | 0.2449 | 0.2513 | 0.1015 | 0.1931 | 0.3079 | 0.2846 | 0.2934 | 0.3315 | 0.2460 | 0.2604 | 0.0868 | 0.2136 | 0.2236 |
| | Mesorch(Zhu et al., 2024) | 0.1855 | 0.2253 | 0.2192 | 0.2128 | 0.1836 | 0.1339 | 0.2225 | 0.2327 | 0.1163 | 0.1744 | 0.2729 | 0.2698 | 0.2775 | 0.2962 | 0.2074 | 0.2237 | 0.0708 | 0.1929 | 0.2065 |
| | MaskCLIP(Dong et al., 2022) | **0.5638** | **0.2885** | **0.5360** | **0.5128** | **0.3636** | **0.6676** | 0.2215 | **0.5473** | **0.4222** | **0.6942** | **0.4875** | **0.5886** | **0.6397** | **0.6737** | **0.5322** | **0.6535** | **0.4492** | **0.5359** | **0.5210** |

## A.2 PROTOCOL-2: CROSS-EDIT-TYPES EVALUATION (CROSS-ET)

Table 17: **Protocol-2:** Cross-ET. Pixel-level (localization) **IoU** performance on Protocol-2.

| | | Rem. (IoU) | | | | | | Rep. (IoU) | | | | | | | | | | | Add. (IoU) | | | Avg (IoU) |
|---|---|---|---|---|---|---|---|---|---|---|---|---|---|---|---|---|---|---|---|---|---|---|---|
| | Model | SD2-Inp | InpAny | ZONE | CLIPW | FLUX-Inp | RORem | BlenDif | PbE | Any4-Inp | Dres-Inp | SDXL-Inp | UltraE | Difree | Kol-Inp | SD3-Conc | SDI.5-Inp | HDPaint | BlenLaDif | PowerP | ACE+ | IoU |
| Rem. | MVSS-Net(Dong et al., 2021) | 0.0823 | 0.1693 | 0.1747 | 0.1395 | 0.1335 | 0.1304 | 0.1072 | 0.1543 | 0.1115 | 0.1148 | 0.1153 | 0.1343 | 0.0727 | 0.0993 | 0.1208 | 0.1135 | 0.1287 | 0.0397 | 0.0717 | 0.0541 | 0.134 |
| | PSCC-Net(Liu et al., 2021) | 0.0445 | 0.0808 | 0.0614 | 0.0568 | 0.0606 | 0.0560 | 0.0662 | 0.0563 | 0.0634 | 0.0568 | 0.0678 | 0.0554 | 0.0580 | 0.0602 | 0.0533 | 0.0486 | 0.0652 | 0.0064 | 0.0087 | 0.0056 | 0.0500 |
| | TruFor(Guillaro et al., 2022) | 0.0661 | 0.1987 | 0.1367 | 0.1562 | 0.1293 | 0.1376 | 0.1692 | 0.1730 | 0.1267 | 0.1177 | 0.0999 | 0.1272 | 0.0412 | 0.0870 | 0.1190 | 0.1098 | 0.1583 | 0.0633 | 0.1150 | 0.0536 | 0.1201 |
| | IML-ViT(Ma et al., 2023) | 0.0854 | 0.2666 | 0.1917 | 0.1928 | 0.2182 | 0.2094 | 0.0912 | 0.2462 | 0.1192 | 0.1204 | 0.1206 | 0.1260 | | 0.0430 | 0.0848 | 0.1104 | 0.1520 | 0.0871 | 0.1075 | 0.0383 | 0.1326 |
| | Mesorch(Zhu et al., 2024) | 0.0004 | 0.0786 | 0.0045 | 0.0087 | 0.0004 | 0.0003 | 0.0011 | 0.0013 | 0.0020 | 0.0007 | 0.0008 | 0.0009 | 0.0009 | 0.0005 | 0.0004 | 0.0016 | 0.0003 | 0.0016 | 0.0004 | 0.0002 | 0.0053 |
| | MaskCLIP(Dong et al., 2022) | **0.1633** | **0.4097** | **0.3880** | **0.3751** | **0.3313** | **0.3061** | **0.4484** | **0.3846** | **0.2537** | **0.2563** | **0.2067** | **0.2353** | **0.1628** | **0.1520** | **0.2789** | **0.2298** | **0.3213** | **0.1932** | **0.3484** | **0.1235** | **0.2784** |
| Rep. | MVSS-Net(Dong et al., 2021) | 0.0745 | 0.0022 | 0.0559 | 0.1351 | 0.0690 | 0.0040 | 0.0243 | 0.2725 | 0.2193 | 0.2155 | 0.1995 | 0.1897 | 0.1418 | 0.1902 | 0.2362 | 0.1631 | 0.3225 | 0.0814 | 0.1743 | 0.0780 | 0.1427 |
| | PSCC-Net(Liu et al., 2021) | 0.0933 | 0.0019 | 0.0754 | 0.1118 | 0.0450 | 0.0118 | 0.3289 | 0.1862 | 0.1642 | 0.1629 | 0.1838 | 0.1171 | 0.1070 | 0.1712 | 0.2267 | 0.1391 | 0.1658 | 0.0480 | 0.1118 | 0.0451 | 0.1249 |
| | TruFor(Guillaro et al., 2022) | 0.0472 | 0.0003 | 0.0293 | 0.1228 | 0.0434 | 0.0011 | 0.3522 | 0.2204 | 0.1869 | 0.1706 | 0.1730 | 0.2175 | 0.0589 | 0.1549 | 0.1632 | 0.1259 | 0.1979 | 0.0689 | 0.1634 | 0.0589 | 0.1278 |
| | IML-ViT(Ma et al., 2023) | 0.087 | 0.0025 | 0.0604 | 0.1401 | 0.0975 | 0.0069 | 0.3739 | 0.2616 | 0.2033 | 0.2036 | 0.2126 | 0.2413 | 0.1105 | 0.2099 | 0.2418 | 0.1609 | 0.2016 | 0.0767 | 0.1704 | 0.0957 | 0.1579 |
| | Mesorch(Zhu et al., 2024) | 0.0585 | 0.0007 | 0.0392 | 0.1213 | 0.0666 | 0.0038 | 0.3299 | 0.2076 | 0.1736 | 0.1692 | 0.1720 | 0.2183 | 0.1003 | 0.1649 | 0.2092 | 0.1297 | 0.1761 | 0.0673 | 0.1158 | 0.0669 | 0.1295 |
| | MaskCLIP(Dong et al., 2022) | **0.1821** | **0.0300** | **0.1533** | **0.3443** | **0.1570** | **0.0215** | **0.7524** | **0.6031** | **0.5200** | **0.4771** | **0.4594** | **0.5822** | **0.4567** | **0.4703** | **0.5531** | **0.3615** | **0.4733** | **0.2729** | **0.5427** | **0.2779** | **0.3845** |
| Add. | MVSS-Net(Dong et al., 2021) | 0.0088 | 0.0008 | 0.0066 | 0.0177 | 0.0201 | 0.0008 | 0.0084 | 0.0407 | 0.0295 | 0.0331 | 0.0217 | 0.0365 | 0.0350 | 0.0287 | 0.0361 | 0.0181 | 0.0339 | 0.1809 | 0.3182 | 0.2799 | 0.0578 |
| | PSCC-Net(Liu et al., 2021) | **0.0225** | 0.0012 | 0.0088 | 0.0390 | 0.0250 | 0.0023 | 0.0161 | 0.0752 | 0.0503 | 0.0554 | 0.0356 | 0.0450 | 0.0527 | 0.0473 | 0.0499 | **0.0362** | 0.0617 | 0.1669 | 0.2657 | 0.2503 | 0.0654 |
| | TruFor(Guillaro et al., 2022) | 0.0071 | 0.0004 | 0.0055 | 0.0186 | 0.0132 | 0.0002 | 0.0056 | 0.0476 | 0.0358 | 0.0384 | 0.0213 | 0.0395 | 0.0255 | 0.0320 | 0.0358 | 0.0175 | 0.0446 | 0.1659 | 0.3444 | 0.2896 | 0.0594 |
| | IML-ViT(Ma et al., 2023) | 0.0036 | 0.0001 | 0.0071 | 0.0074 | 0.0032 | 0.0005 | 0.0022 | 0.0383 | 0.0137 | 0.0237 | 0.0053 | 0.0106 | 0.0148 | 0.0075 | 0.0168 | 0.0083 | 0.0215 | 0.2398 | 0.3826 | 0.0820 | 0.0445 |
| | Mesorch(Zhu et al., 2024) | 0.0027 | 0.0019 | 0.0019 | 0.0085 | 0.0082 | 0.0005 | 0.0043 | 0.0218 | 0.0139 | 0.0184 | 0.0092 | 0.0137 | 0.0186 | 0.0136 | 0.0134 | 0.0096 | 0.0196 | 0.2233 | 0.3529 | 0.3040 | 0.0529 |
| | MaskCLIP(Dong et al., 2022) | **0.0215** | **0.0034** | **0.0099** | **0.0545** | **0.0368** | **0.0024** | **0.0273** | **0.1184** | **0.0825** | **0.0794** | **0.0410** | **0.0836** | **0.0861** | **0.0554** | **0.0719** | 0.0338 | **0.0958** | **0.5688** | **0.7998** | **0.7016** | **0.1487** |

Table 18: **Protocol-2:** Cross-ET. Pixel-level (localization) **F1** performance on Protocol-2.

| | | Rem. (F1) | | | | | | Rep. (F1) | | | | | | | | | | | Add. (F1) | | | Avg (F1) |
|---|---|---|---|---|---|---|---|---|---|---|---|---|---|---|---|---|---|---|---|---|---|---|---|
| | Model | SD2-Inp | InpAny | ZONE | CLIPW | FLUX-Inp | RORem | BlenDif | PbE | Any4-Inp | Dres-Inp | SDXL-Inp | UltraE | Difree | Kol-Inp | SD3-Conc | SDI.5-Inp | HDPaint | BlenLaDif | PowerP | ACE+ | F1 |
| Rem. | MVSS-Net(Dong et al., 2021) | 0.1146 | 0.2088 | 0.2202 | 0.1853 | 0.1778 | 0.1690 | 0.1445 | 0.2047 | 0.1506 | 0.1527 | 0.1533 | 0.1783 | 0.1030 | 0.1335 | 0.1620 | 0.1551 | 0.1707 | 0.0611 | 0.1090 | 0.0795 | 0.1517 |
| | PSCC-Net(Liu et al., 2021) | 0.0689 | 0.1181 | 0.0939 | 0.0849 | 0.0911 | 0.0853 | 0.0985 | 0.0859 | 0.0958 | 0.0875 | 0.1033 | 0.0857 | 0.0432 | 0.0914 | 0.0822 | 0.0756 | 0.0991 | 0.0111 | 0.0151 | 0.0099 | 0.0763 |
| | TruFor(Guillaro et al., 2022) | 0.0867 | 0.2381 | 0.1744 | 0.1959 | 0.1622 | 0.1731 | 0.2071 | 0.2165 | 0.1593 | 0.1473 | 0.1266 | 0.1604 | 0.0764 | 0.1088 | 0.1505 | 0.1401 | 0.1962 | 0.0852 | 0.15343 | 0.0689 | 0.1514 |
| | IML-ViT(Ma et al., 2023) | 0.1095 | 0.3075 | 0.2325 | 0.2325 | 0.2572 | 0.2456 | 0.1076 | 0.2472 | 0.1110 | 0.1465 | 0.1515 | 0.1534 | 0.0017 | 0.0530 | 0.1056 | 0.1110 | 0.1888 | 0.1122 | 0.1388 | 0.0478 | 0.1557 |
| | Mesorch(Zhu et al., 2024) | 0.0008 | 0.0927 | 0.0078 | 0.0135 | 0.0007 | 0.0006 | 0.0019 | 0.0023 | 0.0034 | 0.0013 | 0.0015 | 0.0014 | 0.0017 | 0.0009 | 0.0007 | 0.0029 | 0.0005 | 0.0026 | 0.0007 | 0.0003 | 0.0069 |
| | MaskCLIP(Dong et al., 2022) | **0.2071** | **0.4863** | **0.4845** | **0.4545** | **0.3998** | **0.3772** | **0.5224** | **0.4684** | **0.3075** | **0.3124** | **0.2556** | **0.2855** | **0.2166** | **0.1826** | **0.3450** | **0.2850** | **0.3915** | **0.2477** | **0.4381** | **0.1558** | **0.3411** |
| Rep. | MVSS-Net(Dong et al., 2021) | 0.0951 | 0.0034 | 0.0727 | 0.1367 | 0.0853 | 0.0063 | 0.0262 | 0.3236 | 0.2324 | 0.2155 | 0.2368 | 0.2285 | 0.1475 | 0.2219 | 0.2776 | 0.1994 | 0.3869 | 0.1025 | 0.2236 | 0.0991 | 0.1661 |
| | PSCC-Net(Liu et al., 2021) | 0.1281 | 0.0032 | 0.1061 | 0.1562 | 0.0599 | 0.0199 | 0.3698 | 0.2425 | 0.2149 | 0.2119 | 0.2352 | 0.1489 | 0.1486 | 0.2199 | 0.2847 | 0.1870 | 0.2162 | 0.0711 | 0.1562 | 0.0586 | 0.1619 |
| | TruFor(Guillaro et al., 2022) | 0.0589 | 0.0006 | 0.0378 | 0.1485 | 0.0526 | 0.0017 | 0.3821 | 0.2595 | 0.2208 | 0.2004 | 0.2033 | 0.2538 | 0.0726 | 0.1776 | 0.1922 | 0.1520 | 0.2328 | 0.0841 | 0.2011 | 0.0709 | 0.1502 |
| | IML-ViT(Ma et al., 2023) | 0.1124 | 0.0043 | 0.0810 | 0.1763 | 0.1234 | 0.0116 | 0.4034 | 0.3160 | 0.2503 | 0.2477 | 0.2562 | 0.2903 | 0.1425 | 0.2505 | 0.2863 | 0.2014 | 0.2476 | 0.1037 | 0.2231 | 0.1239 | 0.1926 |
| | Mesorch(Zhu et al., 2024) | 0.0809 | 0.0012 | 0.0558 | 0.1593 | 0.0908 | 0.0064 | 0.3736 | 0.2661 | 0.2211 | 0.2138 | 0.2182 | 0.2746 | 0.1343 | 0.2080 | 0.2595 | 0.1701 | 0.2239 | 0.0933 | 0.1622 | 0.0928 | 0.1653 |
| | MaskCLIP(Dong et al., 2022) | **0.2279** | **0.0417** | **0.2089** | **0.4141** | **0.1919** | **0.0324** | **0.8139** | **0.7035** | **0.6045** | **0.5572** | **0.5389** | **0.6627** | **0.5522** | **0.5286** | **0.6352** | **0.4347** | **0.5599** | **0.3246** | **0.6409** | **0.3354** | **0.4505** |
| Add. | MVSS-Net(Dong et al., 2021) | 0.0125 | 0.0015 | 0.0092 | 0.0241 | 0.026 | 0.0014 | 0.0107 | 0.0536 | 0.0385 | 0.0438 | 0.0306 | 0.474 | 0.0444 | 0.0365 | 0.0475 | 0.0245 | 0.0453 | 0.2089 | 0.3635 | 0.3193 | 0.0908 |
| | PSCC-Net(Liu et al., 2021) | **0.0334** | 0.0022 | 0.0135 | 0.0553 | 0.0342 | 0.0039 | 0.0222 | 0.1021 | 0.0688 | 0.0768 | 0.0511 | 0.0610 | 0.0702 | 0.0628 | 0.0686 | **0.0514** | 0.0848 | 0.2129 | 0.3290 | 0.3037 | 0.0854 |
| | TruFor(Guillaro et al., 2022) | 0.0105 | 0.0007 | 0.0079 | 0.0257 | 0.0183 | 0.0005 | 0.0076 | 0.0646 | 0.0482 | 0.0514 | 0.0313 | 0.0525 | 0.034 | 0.0409 | 0.0484 | 0.024 | 0.0611 | 0.1876 | 0.3818 | 0.3234 | 0.0710 |
| | IML-ViT(Ma et al., 2023) | 0.0055 | 0.0002 | 0.0098 | 0.0101 | 0.0044 | 0.0007 | 0.0030 | 0.0521 | 0.0186 | 0.0318 | 0.0077 | 0.01480 | 0.0205 | 0.0104 | 0.0222 | 0.0114 | 0.0303 | 0.2715 | 0.4147 | 0.096 | 0.0518 |
| | Mesorch(Zhu et al., 2024) | 0.0039 | 0.0004 | 0.0027 | 0.0116 | 0.0107 | 0.0008 | 0.0060 | 0.0293 | 0.0083 | 0.0237 | 0.0136 | 0.0180 | 0.0243 | 0.0168 | 0.0179 | 0.0127 | 0.0260 | 0.2506 | 0.3891 | 0.3372 | 0.0607 |
| | MaskCLIP(Dong et al., 2022) | **0.0320** | **0.0054** | **0.0140** | **0.0755** | **0.0490** | **0.0040** | **0.0361** | **0.1570** | **0.1103** | **0.1058** | **0.0596** | **0.1129** | **0.1152** | **0.0723** | **0.0963** | **0.0483** | **0.1304** | **0.6410** | **0.8614** | **0.7619** | **0.1744** |

## A.3 PROTOCOL-3: CROSS-SEMANTIC-LABELS EVALUATION (CROSS-SL)

Table 19: **Protocol-3: Addition**. Pixel-level (localization) performance.

| | | Human | | Animal | | Object | | Avg | |
|---|---|---|---|---|---|---|---|---|---|
| | | IoU | F1 | IoU | F1 | IoU | F1 | IoU | F1 |
| **Animal** | MVSS - Net(Dong et al., 2021) | 0.1282 | 0.1613 | 0.2584 | 0.2991 | 0.1055 | 0.1319 | 0.1640 | 0.1974 |
| | PSCC - Net(Liu et al., 2021) | 0.1395 | 0.1922 | 0.1977 | 0.262 | 0.1258 | 0.1668 | 0.1543 | 0.2070 |
| | TruFor(Guillaro et al., 2022) | 0.0443 | 0.0605 | 0.2191 | 0.2668 | 0.0558 | 0.0731 | 0.1064 | 0.1335 |
| | IML - ViT(Ma et al., 2023) | 0.1282 | 0.1613 | 0.2584 | 0.2991 | 0.1055 | 0.1319 | 0.1640 | 0.1974 |
| | Mesorch(Zhu et al., 2024) | 0.1466 | 0.1768 | 0.2831 | 0.3209 | 0.1018 | 0.1236 | 0.1772 | 0.2071 |
| | MaskCLIP(Dong et al., 2022) | **0.3552** | **0.4151** | **0.7386** | **0.7984** | **0.3803** | **0.4356** | **0.4914** | **0.5497** |
| **Object** | MVSS - Net(Dong et al., 2021) | 0.0976 | 0.1416 | 0.0820 | 0.1256 | 0.0939 | 0.1396 | 0.0912 | 0.1356 |
| | PSCC - Net(Liu et al., 2021) | 0.1479 | 0.2034 | 0.1556 | 0.2144 | 0.1725 | 0.2282 | 0.1587 | 0.2153 |
| | TruFor(Guillaro et al., 2022) | 0.0621 | 0.0836 | 0.078 | 0.1059 | 0.1281 | 0.1619 | 0.0894 | 0.1171 |
| | IML - ViT(Ma et al., 2023) | 0.1886 | 0.2262 | 0.2246 | 0.2662 | 0.2199 | 0.2567 | 0.2110 | 0.2497 |
| | Mesorch(Zhu et al., 2024) | 0.1154 | 0.1466 | 0.1074 | 0.1427 | 0.1442 | 0.1798 | 0.1223 | 0.1564 |
| | MaskCLIP(Dong et al., 2022) | **0.5427** | **0.6155** | **0.6509** | **0.7237** | **0.6228** | **0.6956** | **0.6055** | **0.6783** |

Table 20: **Protocol-3: Removal**. Pixel-level (localization) performance.

| | | Human | | Animal | | Object | | Avg | |
|---|---|---|---|---|---|---|---|---|---|
| | | IoU | F1 | IoU | F1 | IoU | F1 | IoU | F1 |
| **Human** | MVSS-Net(Dong et al., 2021) | 0.0894 | 0.1321 | 0.1029 | 0.1465 | 0.0800 | 0.1145 | 0.0908 | 0.1310 |
| | PSCC-Net(Liu et al., 2021) | 0.0795 | 0.1172 | 0.0877 | 0.1264 | 0.0661 | 0.0946 | 0.0778 | 0.1127 |
| | TruFor(Guillaro et al., 2022) | 0.0276 | 0.0393 | 0.0236 | 0.0337 | 0.0227 | 0.032 | 0.0247 | 0.0350 |
| | IML-ViT(Ma et al., 2023) | 0.0529 | 0.0761 | 0.0478 | 0.0683 | 0.0395 | 0.056 | 0.0467 | 0.0668 |
| | Mesorch(Zhu et al., 2024) | 0.0021 | 0.0033 | 0.0031 | 0.0048 | 0.0023 | 0.0035 | 0.0025 | 0.0039 |
| | MaskCLIP(Dong et al., 2022) | **0.2411** | **0.3142** | **0.2184** | **0.2815** | **0.1639** | **0.2082** | **0.2078** | **0.2679** |
| **Animal** | MVSS-Net(Dong et al., 2021) | **0.0696** | **0.1054** | **0.1268** | **0.1771** | **0.0815** | **0.1152** | **0.0926** | **0.1325** |
| | PSCC-Net(Liu et al., 2021) | 0.0520 | 0.0780 | 0.1105 | 0.1569 | 0.0655 | 0.0924 | 0.0760 | 0.1091 |
| | TruFor(Guillaro et al., 2022) | 0.0032 | 0.0055 | 0.0106 | 0.0164 | 0.0042 | 0.0067 | 0.006 | 0.0095 |
| | IML-ViT(Ma et al., 2023) | 0.0073 | 0.012 | 0.0107 | 0.0168 | 0.0076 | 0.0117 | 0.0085 | 0.0135 |
| | Mesorch(Zhu et al., 2024) | 0.0010 | 0.0017 | 0.0040 | 0.0057 | 0.0019 | 0.0028 | 0.0023 | 0.0034 |
| | MaskCLIP(Dong et al., 2022) | 0.014 | 0.022 | 0.024 | 0.0354 | 0.0154 | 0.0219 | 0.0178 | 0.0264 |
| **Object** | MVSS-Net(Dong et al., 2021) | 0.0648 | 0.0972 | 0.1018 | 0.1425 | 0.0916 | 0.1265 | 0.0861 | 0.1221 |
| | PSCC-Net(Liu et al., 2021) | 0.0580 | 0.0853 | 0.0875 | 0.1233 | 0.0802 | 0.1112 | 0.0752 | 0.1066 |
| | TruFor(Guillaro et al., 2022) | 0.0147 | 0.0221 | 0.0342 | 0.0476 | 0.0424 | 0.0573 | 0.0304 | 0.0423 |
| | IML-ViT(Ma et al., 2023) | 0.0294 | 0.0419 | 0.0471 | 0.0635 | 0.0513 | 0.0672 | 0.0426 | 0.0575 |
| | Mesorch(Zhu et al., 2024) | 0.0042 | 0.0065 | 0.0151 | 0.0214 | 0.0175 | 0.0237 | 0.0123 | 0.0172 |
| | MaskCLIP(Dong et al., 2022) | **0.1420** | **0.1907** | **0.2149** | **0.2742** | **0.2301** | **0.2859** | **0.1957** | **0.2503** |

Table 21: **Protocol-3: Replacement.** Pixel-level (localization) performance.

| | | Human | | Animal | | Object | | Avg | |
|---|---|---|---|---|---|---|---|---|---|
| | | IoU | F1 | IoU | F1 | IoU | F1 | IoU | F1 |
| **Human** | MVSS-Net(Dong et al., 2021) | 0.1142 | 0.1682 | 0.1146 | **0.1687** | **0.1211** | **0.1710** | 0.1166 | 0.1693 |
| | PSCC-Net(Liu et al., 2021) | 0.1178 | 0.1689 | 0.1015 | 0.1414 | 0.0638 | 0.0885 | 0.0944 | 0.1329 |
| | TruFor(Guillaro et al., 2022) | 0.1265 | 0.1713 | 0.0663 | 0.0925 | 0.0428 | 0.0591 | 0.0785 | 0.1076 |
| | IML-ViT(Ma et al., 2023) | 0.1528 | 0.1996 | 0.1018 | 0.1333 | 0.0566 | 0.0745 | 0.1037 | 0.1358 |
| | Mesorch(Zhu et al., 2024) | 0.1278 | 0.1685 | 0.0543 | 0.0741 | 0.0318 | 0.0433 | 0.0713 | 0.0953 |
| | MaskCLIP(Dong et al., 2022) | **0.3907** | **0.4961** | 0.1223 | 0.1585 | 0.1144 | 0.1451 | **0.2091** | **0.2666** |
| **Animal** | MVSS-Net(Dong et al., 2021) | 0.0879 | 0.1302 | 0.1832 | 0.2401 | 0.0942 | 0.1309 | 0.1218 | 0.1671 |
| | PSCC-Net(Liu et al., 2021) | 0.0930 | 0.1324 | 0.1639 | 0.2192 | 0.0786 | 0.1068 | 0.1118 | 0.1528 |
| | TruFor(Guillaro et al., 2022) | 0.0390 | 0.0539 | 0.1676 | 0.2107 | 0.0362 | 0.0473 | 0.0809 | 0.1040 |
| | IML-ViT(Ma et al., 2023) | 0.0643 | 0.0877 | 0.1841 | 0.2277 | 0.0482 | 0.0616 | 0.0988 | 0.1257 |
| | Mesorch(Zhu et al., 2024) | 0.0284 | 0.0380 | 0.1994 | 0.2418 | 0.0364 | 0.0459 | 0.0881 | 0.1086 |
| | MaskCLIP(Dong et al., 2022) | **0.1845** | **0.2418** | **0.4715** | **0.5698** | **0.1654** | **0.2060** | **0.2738** | **0.3392** |
| **Object** | MVSS-Net(Dong et al., 2021) | 0.0849 | 0.1261 | 0.1235 | 0.1718 | 0.1329 | 0.1784 | 0.1138 | 0.1588 |
| | PSCC-Net(Liu et al., 2021) | 0.1053 | 0.1503 | 0.1238 | 0.1672 | 0.1320 | 0.1750 | 0.1204 | 0.1642 |
| | TruFor(Guillaro et al., 2022) | 0.0221 | 0.0310 | 0.0550 | 0.0723 | 0.1010 | 0.1276 | 0.0594 | 0.0770 |
| | IML-ViT(Ma et al., 2023) | 0.0626 | 0.0852 | 0.0797 | 0.1031 | 0.1079 | 0.1350 | 0.0834 | 0.1078 |
| | Mesorch(Zhu et al., 2024) | 0.0460 | 0.0602 | 0.0532 | 0.0693 | 0.1071 | 0.1313 | 0.0688 | 0.0869 |
| | MaskCLIP(Dong et al., 2022) | **0.2240** | **0.2930** | **0.2515** | **0.3156** | **0.3609** | **0.4327** | **0.2788** | **0.3471** |

## A.4 PROTOCOL-4: CROSS-EDIT-GRANULARITY EVALUATION (CROSS-EG)

Table 22: **Protocol-4**. Pixel-level (localization) performance.

| | Model | Area - 1 | | Area - 2 | | Area - 3 | | Avg | |
|---|---|---|---|---|---|---|---|---|---|
| | | IoU | F1 | IoU | F1 | IoU | F1 | IoU | F1 |
| Area - 1 | MVSS - Net(Dong et al., 2021) | 0.0725 | 0.0991 | 0.0487 | 0.0676 | 0.0130 | 0.0219 | 0.0447 | 0.0629 |
| | PSCC - Net(Liu et al., 2021) | 0.0827 | 0.1172 | 0.0807 | 0.1132 | **0.0309** | **0.0511** | 0.0648 | 0.0938 |
| | TruFor(Guillaro et al., 2022) | 0.0525 | 0.0674 | 0.0199 | 0.0286 | 0.0020 | 0.0034 | 0.0248 | 0.0331 |
| | IML - ViT(Ma et al., 2023) | 0.1208 | 0.1479 | 0.0555 | 0.0737 | 0.0060 | 0.0101 | 0.0608 | 0.0772 |
| | Mesorch(Zhu et al., 2024) | 0.0719 | 0.0923 | 0.0253 | 0.0355 | 0.0017 | 0.0031 | 0.0330 | 0.0436 |
| | MaskCLIP(Dong et al., 2022) | **0.2790** | **0.3439** | **0.1441** | **0.1951** | 0.0121 | 0.0220 | **0.1451** | **0.1870** |
| Area - 2 | MVSS - Net(Dong et al., 2021) | 0.0359 | 0.0584 | 0.1291 | 0.1765 | 0.1156 | 0.1584 | 0.0935 | 0.1311 |
| | PSCC - Net(Liu et al., 2021) | 0.0381 | 0.0620 | 0.1227 | 0.1707 | 0.1358 | 0.1860 | 0.0989 | 0.1396 |
| | TruFor(Guillaro et al., 2022) | 0.0607 | 0.0797 | 0.1404 | 0.1714 | 0.0679 | 0.0932 | 0.0890 | 0.1148 |
| | IML - ViT(Ma et al., 2023) | 0.1049 | 0.1338 | 0.2180 | 0.2591 | 0.1208 | 0.1584 | 0.1479 | 0.1838 |
| | Mesorch(Zhu et al., 2024) | 0.0786 | 0.0992 | 0.2027 | 0.2400 | 0.0964 | 0.1282 | 0.1259 | 0.1558 |
| | MaskCLIP(Dong et al., 2022) | **0.2080** | **0.2675** | **0.4063** | **0.4927** | **0.1981** | **0.2727** | **0.2708** | **0.3443** |
| Area - 3 | MVSS-Net(Dong et al., 2021) | 0.0150 | 0.0266 | 0.0871 | 0.1307 | 0.2685 | 0.3216 | 0.1235 | 0.1596 |
| | PSCC-Net(Liu et al., 2021) | 0.0157 | 0.0271 | 0.0763 | 0.1168 | 0.2473 | 0.3070 | 0.1131 | 0.1503 |
| | TruFor(Guillaro et al., 2022) | 0.0303 | 0.0459 | 0.1239 | 0.1633 | 0.2862 | 0.3338 | 0.1468 | 0.1810 |
| | IML-ViT(Ma et al., 2023) | 0.0440 | 0.0620 | 0.1731 | 0.2124 | 0.3413 | 0.3805 | 0.1861 | 0.2183 |
| | Mesorch(Zhu et al., 2024) | 0.0234 | 0.0354 | 0.1514 | 0.1900 | 0.3456 | 0.3850 | 0.1735 | 0.2035 |
| | MaskCLIP(Dong et al., 2022) | **0.0721** | **0.1050** | **0.3053** | **0.3839** | **0.6328** | **0.7193** | **0.3367** | **0.4027** |

## A.5 PROTOCOL-5: TOWARD-REALWORLD-IMDL EVALUATION (REALWORLD-IMDL)

Table 23: **Protocol-5**. Pixel-level (localization) performance.

| Training | Model | Meitu | | Photoshop | | Avg | |
|---|---|---|---|---|---|---|---|
| | | IoU | F1 | IoU | F1 | IoU | F1 |
| Set - 1 | MVSS - Net(Dong et al., 2021) | 0.2034 | 0.2459 | 0.0380 | 0.0551 | 0.1207 | 0.1505 |
| | PSCC - Net(Liu et al., 2021) | 0.1723 | 0.2249 | 0.0163 | 0.0252 | 0.0943 | 0.1251 |
| | TruFor(Guillaro et al., 2022) | 0.2707 | 0.3232 | 0.0131 | 0.0192 | 0.1419 | 0.1712 |
| | IML - ViT(Ma et al., 2023) | 0.2716 | 0.3236 | 0.0081 | 0.0131 | 0.1399 | 0.1684 |
| | Mesorch(Zhu et al., 2024) | 0.2755 | 0.3214 | 0.0103 | 0.0142 | 0.1429 | 0.1678 |
| | MaskCLIP(Dong et al., 2022) | **0.4795** | **0.5748** | **0.1798** | **0.2400** | **0.3297** | **0.4074** |
| Set - 2 | MVSS - Net(Dong et al., 2021) | 0.0455 | 0.0659 | 0.0525 | 0.0755 | 0.0490 | 0.0707 |
| | PSCC - Net(Liu et al., 2021) | 0.0277 | 0.0423 | 0.0299 | 0.0456 | 0.0288 | 0.0440 |
| | TruFor(Guillaro et al., 2022) | 0.0408 | 0.0567 | 0.0169 | 0.0246 | 0.0289 | 0.0407 |
| | IML - ViT(Ma et al., 2023) | 0.0101 | 0.0145 | 0.0193 | 0.0279 | 0.0147 | 0.0212 |
| | Mesorch(Zhu et al., 2024) | 0.0305 | 0.0425 | 0.0220 | 0.0318 | 0.0263 | 0.0372 |
| | MaskCLIP(Dong et al., 2022) | **0.0824** | **0.1169** | **0.1392** | **0.1944** | **0.1108** | **0.1557** |

Table 24: **Protocol-5**. Image-level (detection) performance.

| Training | Model | Doubao | Gemini | GPT | Avg |
|---|---|---|---|---|---|
| | | Acc | Acc | Acc | Acc |
| Set - 1 | FreqNet(Cai et al., 2021) | 0.4443 | 0.4975 | 0.3956 | 0.4458 |
| | UniFD(Ojha et al., 2023) | **0.7080** | **0.6527** | **0.6474** | **0.6694** |
| | NPR(Tan et al., 2024a) | 0.4863 | 0.5201 | 0.4204 | 0.4756 |
| | AIDE(Yan et al., 2024a) | 0.4873 | 0.5621 | 0.4216 | 0.4903 |
| | FIRE(Chu et al., 2024) | 0.4766 | 0.5050 | 0.5029 | 0.4948 |
| Set - 2 | FreqNet(Cai et al., 2021) | 0.5625 | 0.5805 | 0.4511 | 0.5314 |
| | UniFD(Ojha et al., 2023) | **0.7344** | 0.6745 | **0.6215** | **0.6768** |
| | NPR(Tan et al., 2024a) | 0.5039 | 0.5713 | 0.4723 | 0.5158 |
| | AIDE(Yan et al., 2024a) | 0.5400 | **0.7492** | 0.5136 | 0.6009 |
| | FIRE(Chu et al., 2024) | 0.4561 | 0.4807 | 0.4900 | 0.4756 |
| Set - 3 | FreqNet(Cai et al., 2021) | 0.3750 | 0.3926 | 0.3408 | 0.3695 |
| | UniFD(Ojha et al., 2023) | **0.7363** | **0.7450** | **0.7011** | **0.7275** |
| | NPR(Tan et al., 2024a) | 0.3594 | 0.4883 | 0.2500 | 0.3659 |
| | AIDE(Yan et al., 2024a) | 0.4863 | 0.5570 | 0.4947 | 0.5127 |
| | FIRE(Chu et al., 2024) | 0.5225 | 0.5252 | 0.4953 | 0.5143 |
| Set - 4 | FreqNet(Cai et al., 2021) | 0.3467 | 0.4773 | 0.3986 | 0.4075 |
| | UniFD(Ojha et al., 2023) | **0.7363** | **0.8003** | **0.7488** | **0.7618** |
| | NPR(Tan et al., 2024a) | 0.5098 | 0.6779 | 0.5772 | 0.5883 |
| | AIDE(Yan et al., 2024a) | 0.4717 | 0.3817 | 0.4481 | 0.4338 |
| | FIRE(Chu et al., 2024) | 0.5703 | 0.5914 | 0.5825 | 0.5814 |

## B  MORE DETAILS ABOUT THE DATA

We provide a brief view of the samples generated by different models included in our dataset.

### B.1  MASK-BASED

#### B.1.1  ADDITION

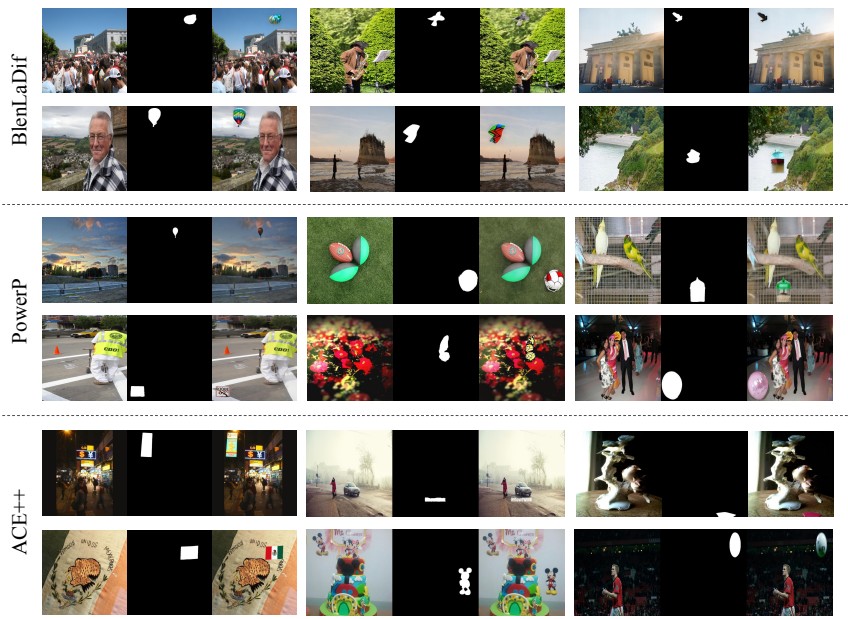

Figure 6: **Addition**. Edit by BlenLaDif (Avrahami et al., 2023); PowerP (Zhuang et al., 2024); ACE++ (Mao et al., 2025)

#### B.1.2  NULL-TEXT

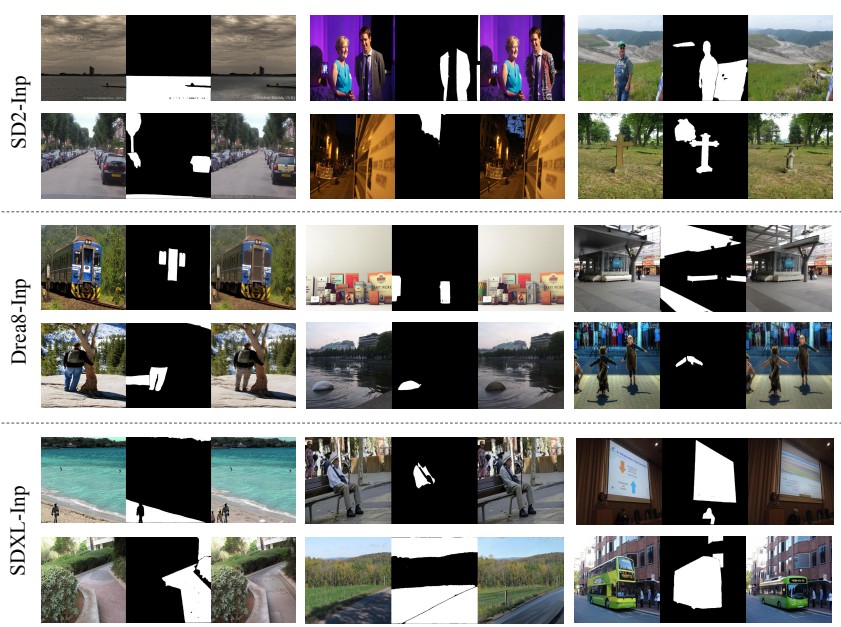

Figure 7: **Null-Text**. Edit by SD2-Inp (AI, 2022); Drea8-Inp (Lykon, 2024); SDXL-Inp (AI, 2023)

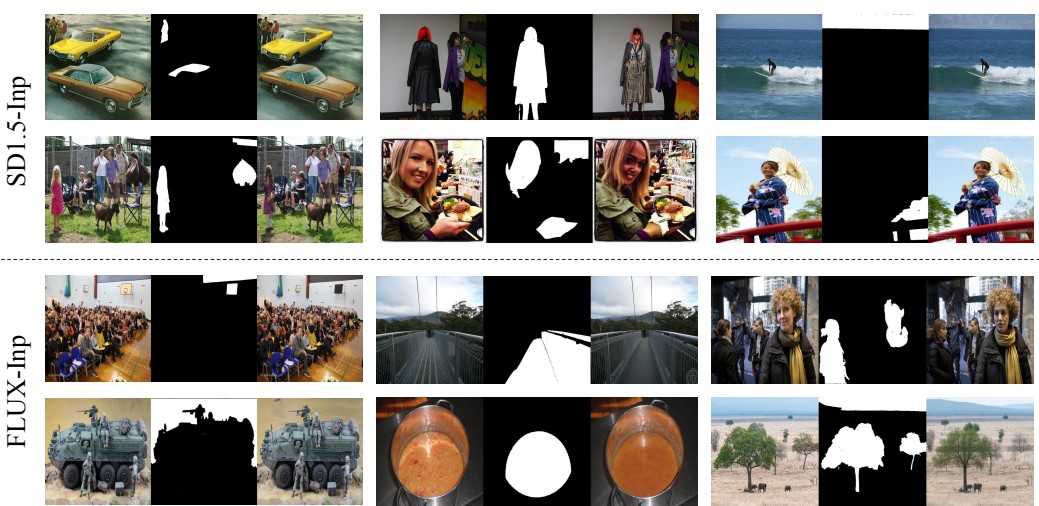

Figure 8: **Null-Text**. Edit by (Rombach & Esser, 2022); Drea8-Inp (alimama creative, 2024a)

### B.1.3 REMOVAL

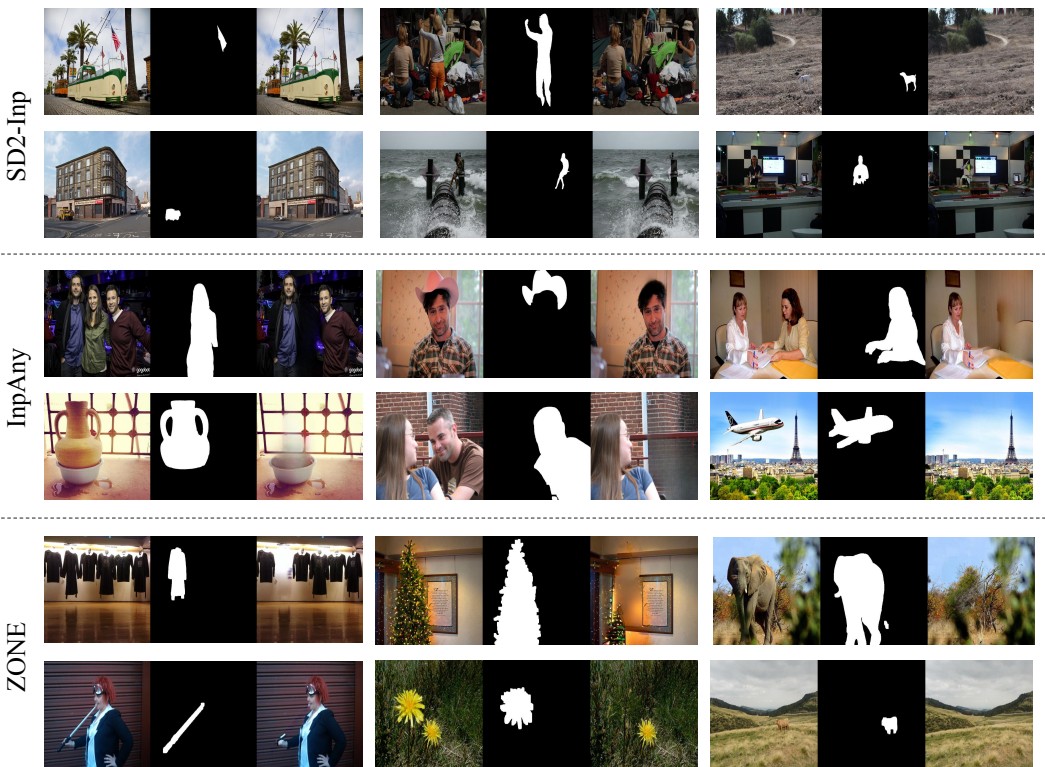

Figure 9: **Removal**. Edit by SD2-Inp (AI, 2022); InpAny (Yu et al., 2023); SDXL-Inp (Li et al., 2024)

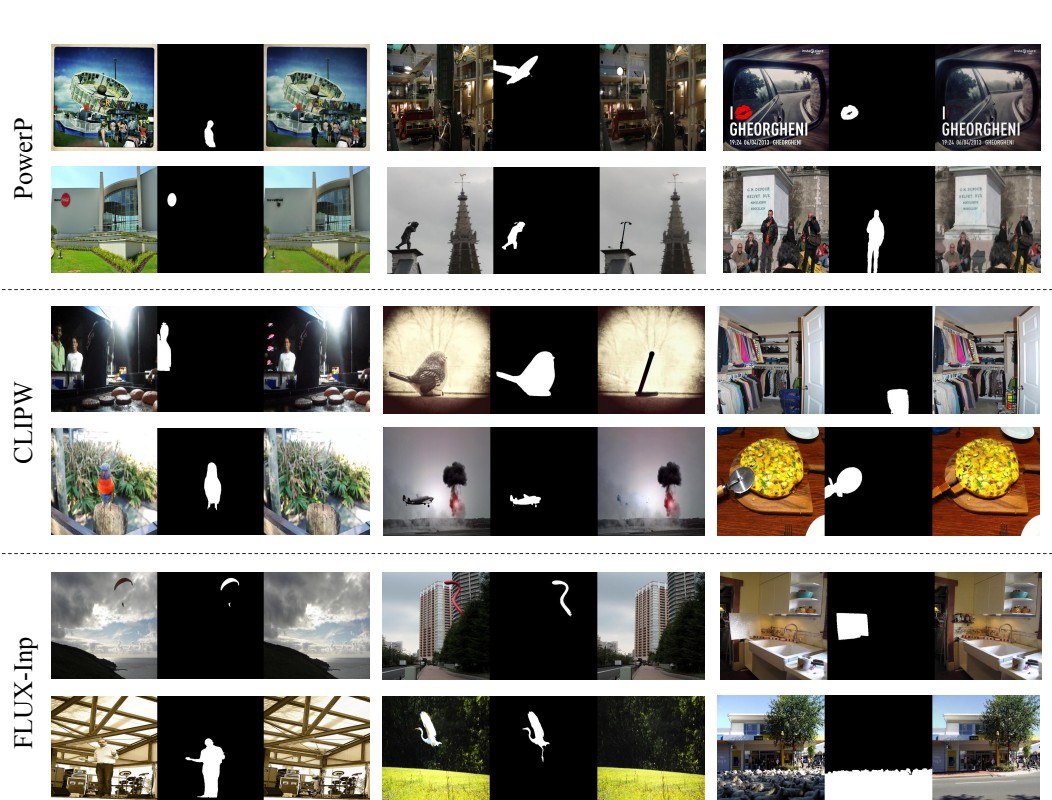

Figure 10: **Removal**. Edit by PowerP (Zhuang et al., 2024); CLIPW (Ekin et al., 2024); FLUX-Inp (alimama creative, 2024a)

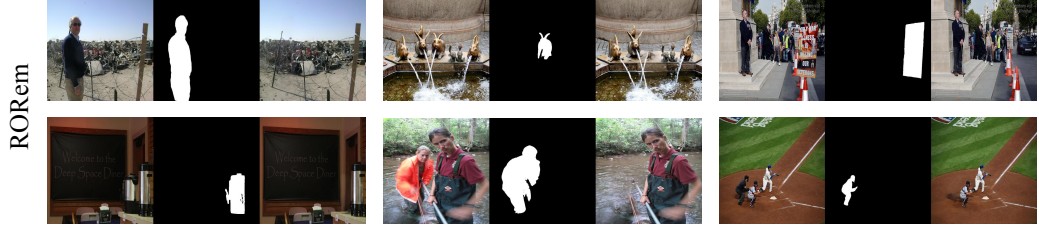

Figure 11: **Removal**. Edit by RORem (Li et al., 2025)

### B.1.4 REPLACEMENT

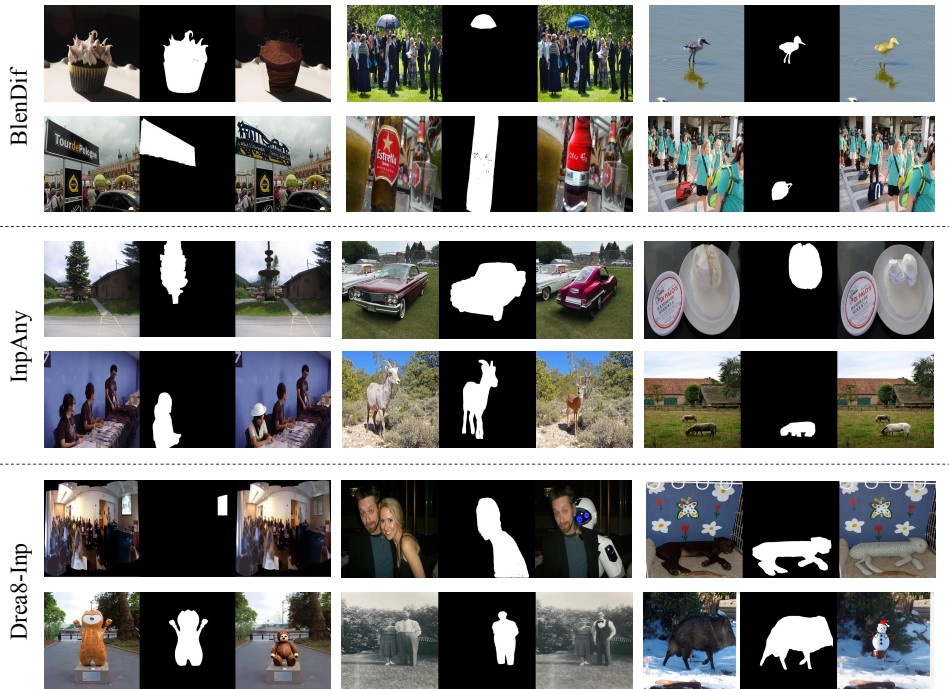

Figure 12: **Replacement**. Edit by BlenDif (Avrahami et al., 2022); InpAny (Yu et al., 2023); Drea8-Inp (Lykon, 2024)

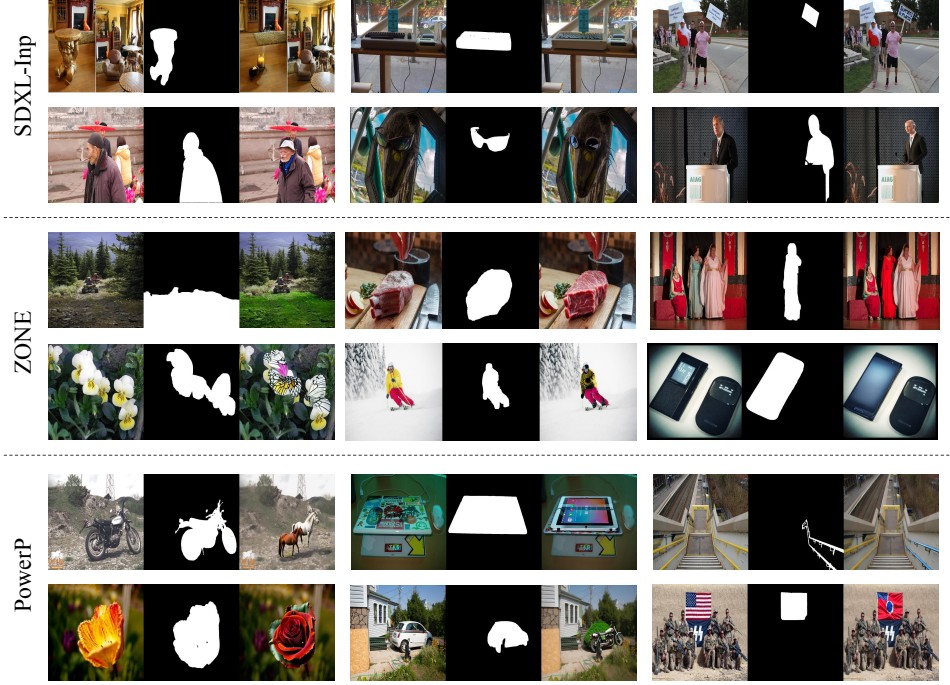

Figure 13: **Replacement**. Edit by SDXL-Inp (AI, 2023); ZONE (Li et al., 2024); PowerP (Zhuang et al., 2024)

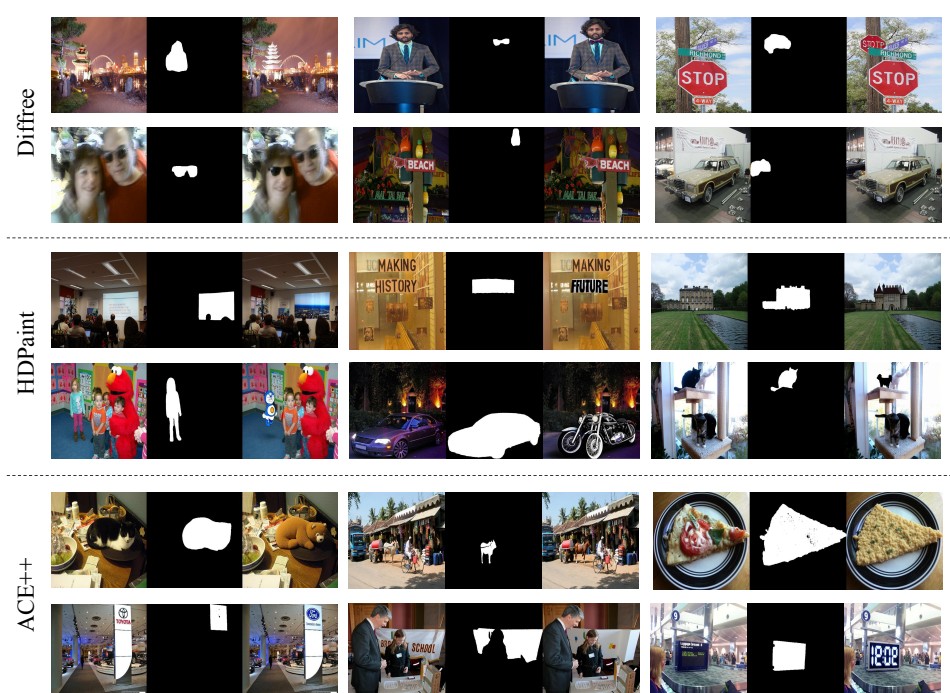

Figure 14: **Replacement**. Edit by Diffree (Zhao et al., 2024b); HDPaint (Manukyan et al., 2023); ACE++ (Mao et al., 2025)

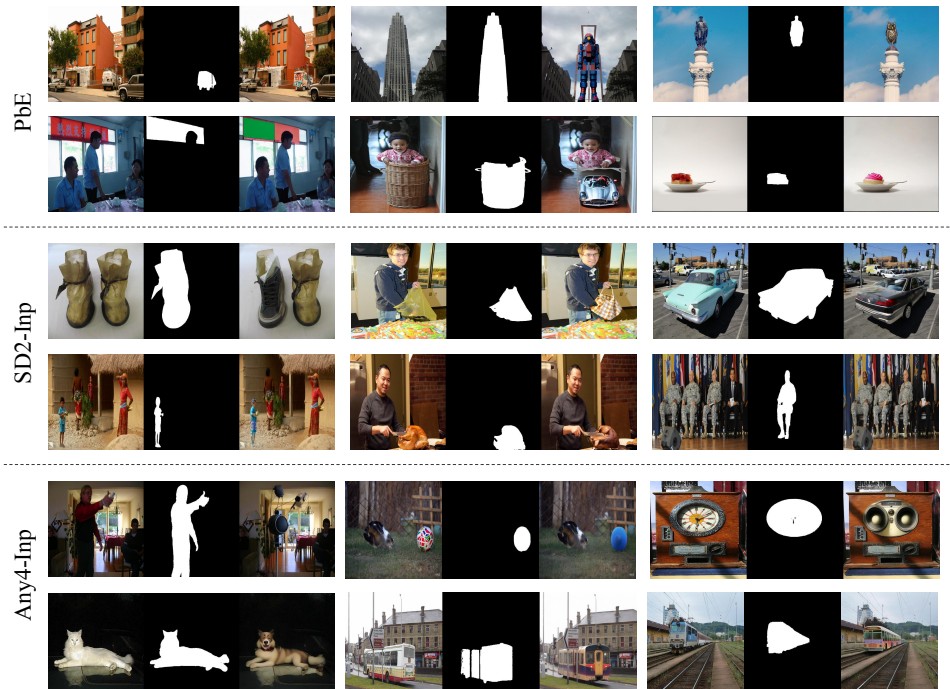

Figure 15: **Replacement**. Edit by PbE (Yang et al., 2023); SD-Inp (AI, 2022); Any4-Inp (Sanster, 2024)

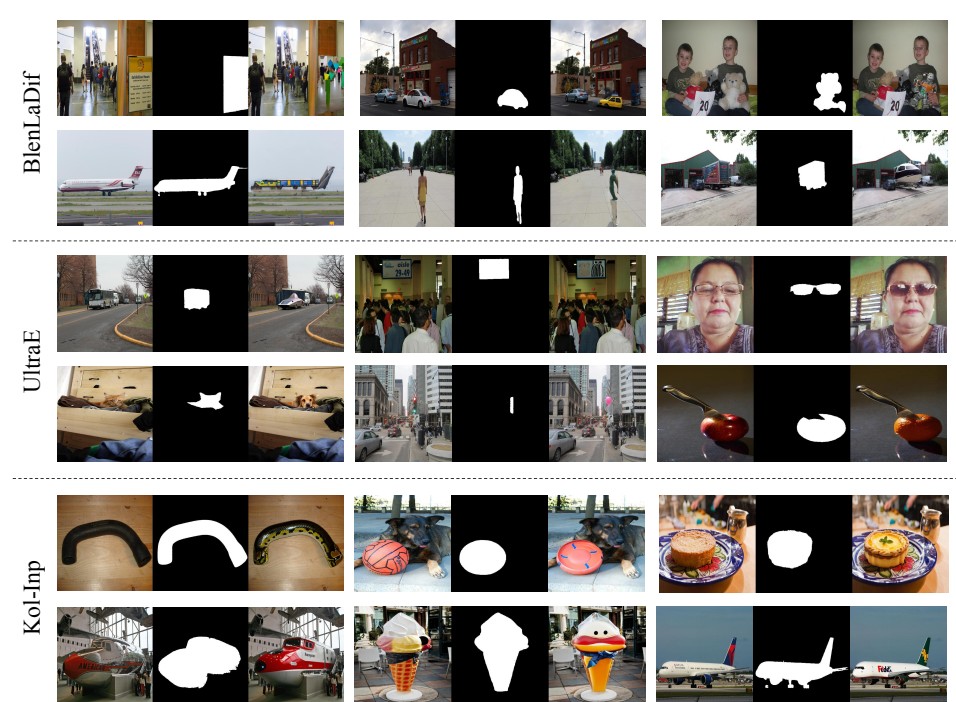

Figure 16: **Replacement**. Edit by BlenLaDif (Avrahami et al., 2023); UltraE (Zhao et al., 2024a); Kol-Inp (Kwai-Kolors, 2024)

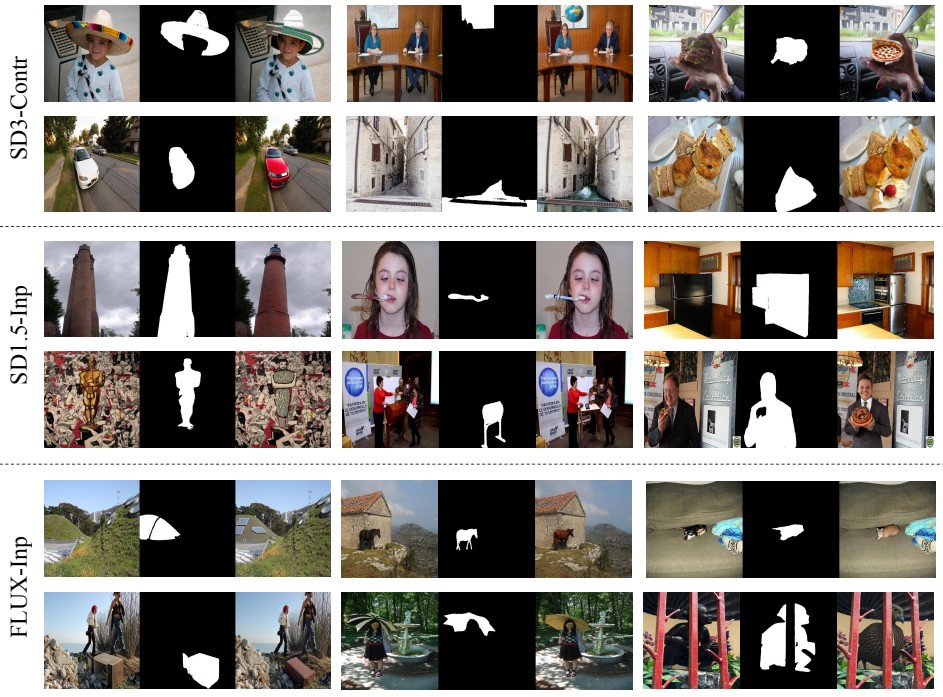

Figure 17: **Replacement**. Edit by SD3-Contr (alimama creative, 2024b); SD1.5-Inp (Rombach & Esser, 2022); FLUX-Inp (alimama creative, 2024a)

## B.2 TEXT-BASED

### B.2.1 ADDITION

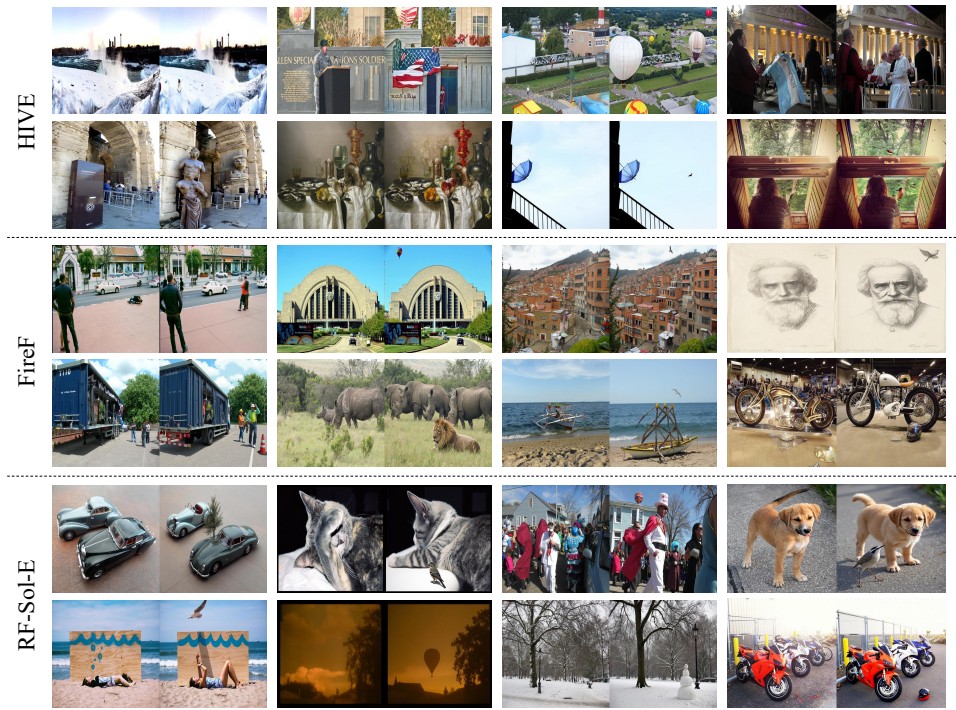

Figure 18: **Addition**. Edit by HIVE (Zhang et al., 2024a); FireF (Deng et al., 2024); RF-Sol-E (Wang et al., 2024a)

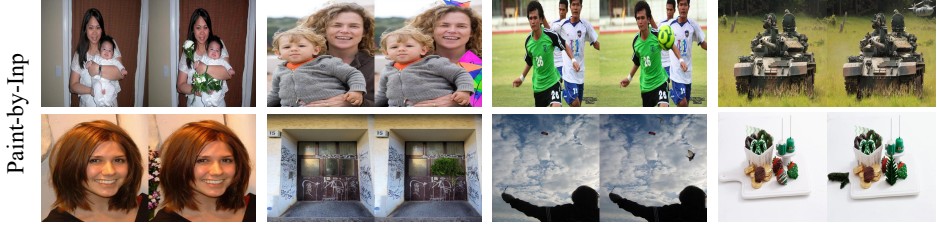

Figure 19: **Addition**. Edit by Paint-by-Inp (Wasserman et al., 2024)

### B.2.2 REMOVAL

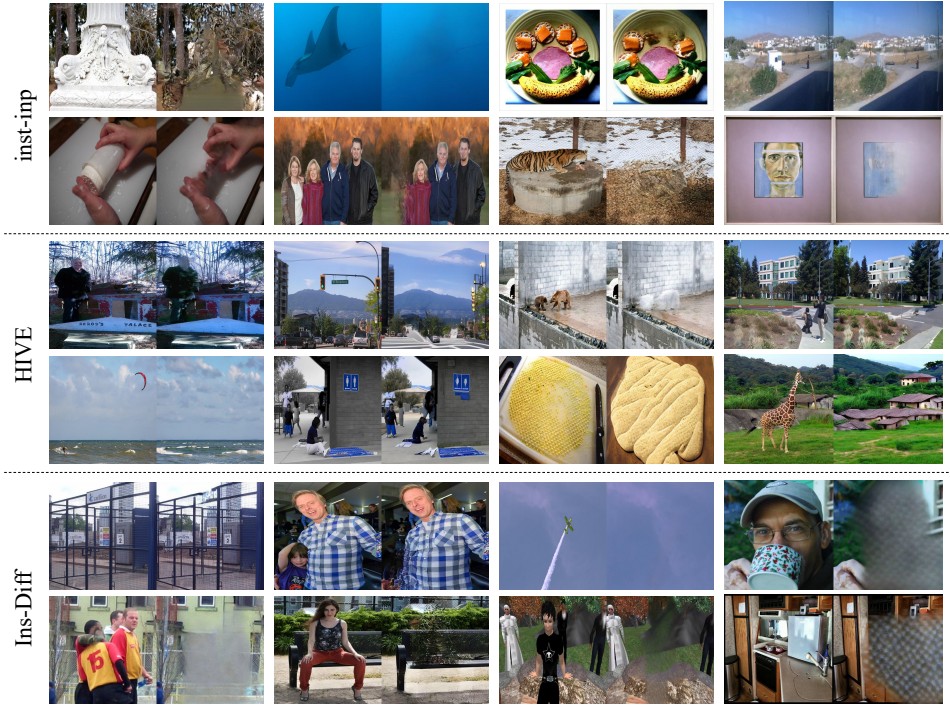

Figure 20: **Removal**. Edit by inst-inp (Yildirim et al., 2023); HIVE (Zhang et al., 2024a), Ins-Diff (Geng et al., 2024)

### B.2.3 REPLACEMENT

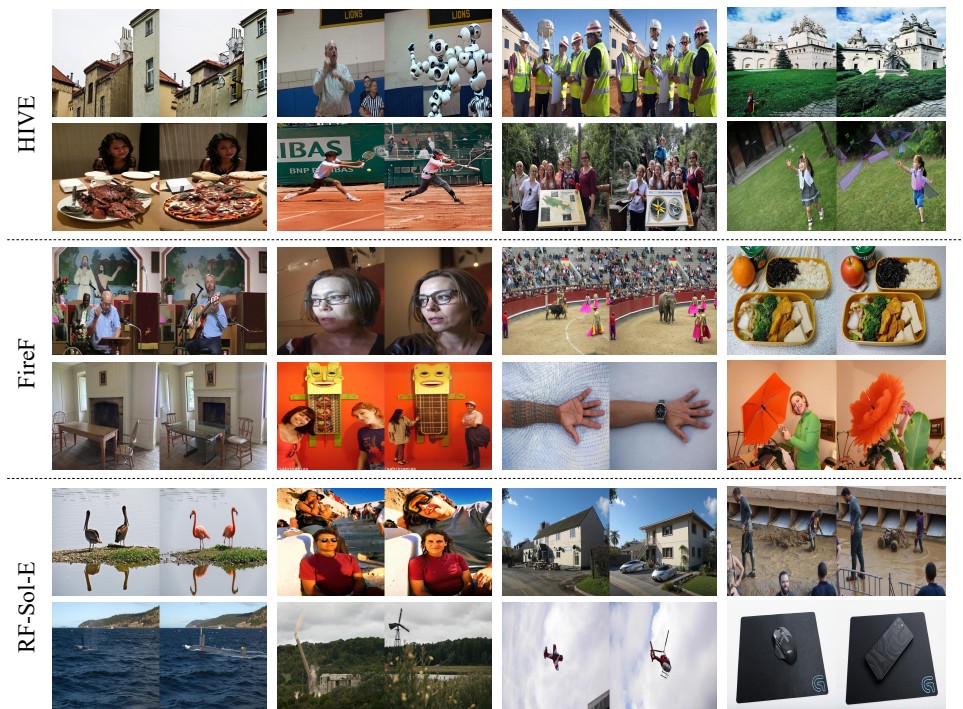

Figure 21: **Replacement**. Edit by HIVE (Zhang et al., 2024a); FireF (Deng et al., 2024); RF-Sol-E (Wang et al., 2024a)

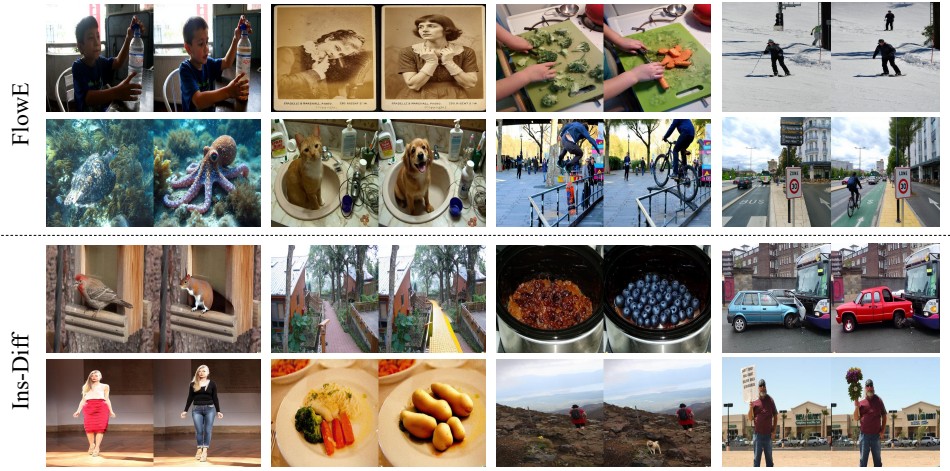

Figure 22: **Replacement**. Edit by FireF (Deng et al., 2024); Ins-Diff

### B.3 COMMERCIAL TOOLS

### B.3.1 MASK-BASED

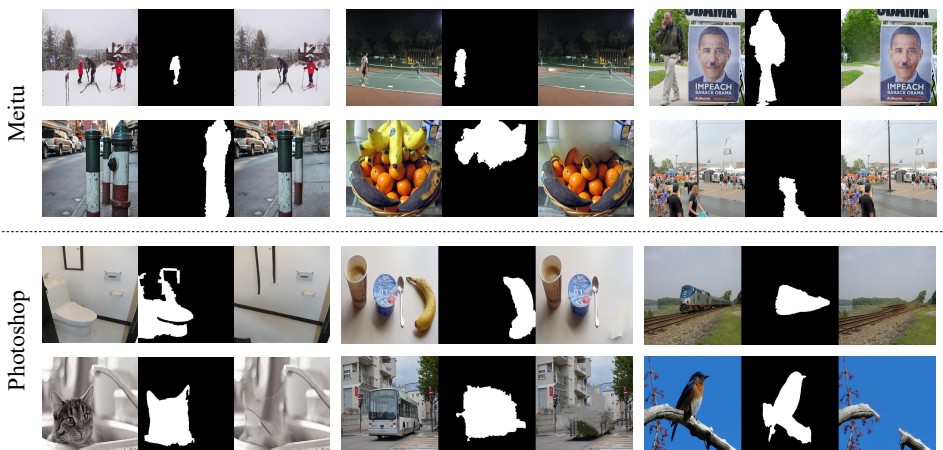

Figure 23: **Commercial Tools**. Edit by Meitu (Meitu Network Technology Co., 2023); Photoshop (Inc., 2023)

### B.3.2 TEXT-BASED

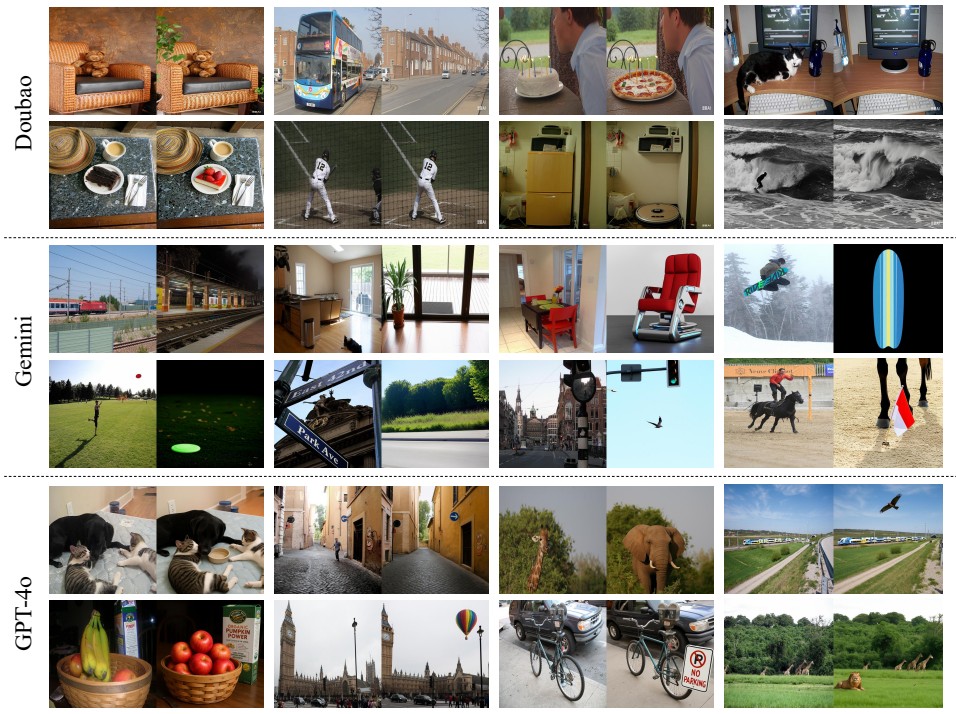

Figure 24: **Commercial Tools**. Edit by Doubao (ByteDance, 2023); Gemini (Google DeepMind Team); GPT-4o (OpenAI, 2025)

## C VISUALIZATION OF IMDL MODELS' OUTPUT

We include visualization of some of the IMDL models' prediction results in Protocol-2 to provide a qualitative overview.

### C.1 MODELS TRAINED ON ADDITION

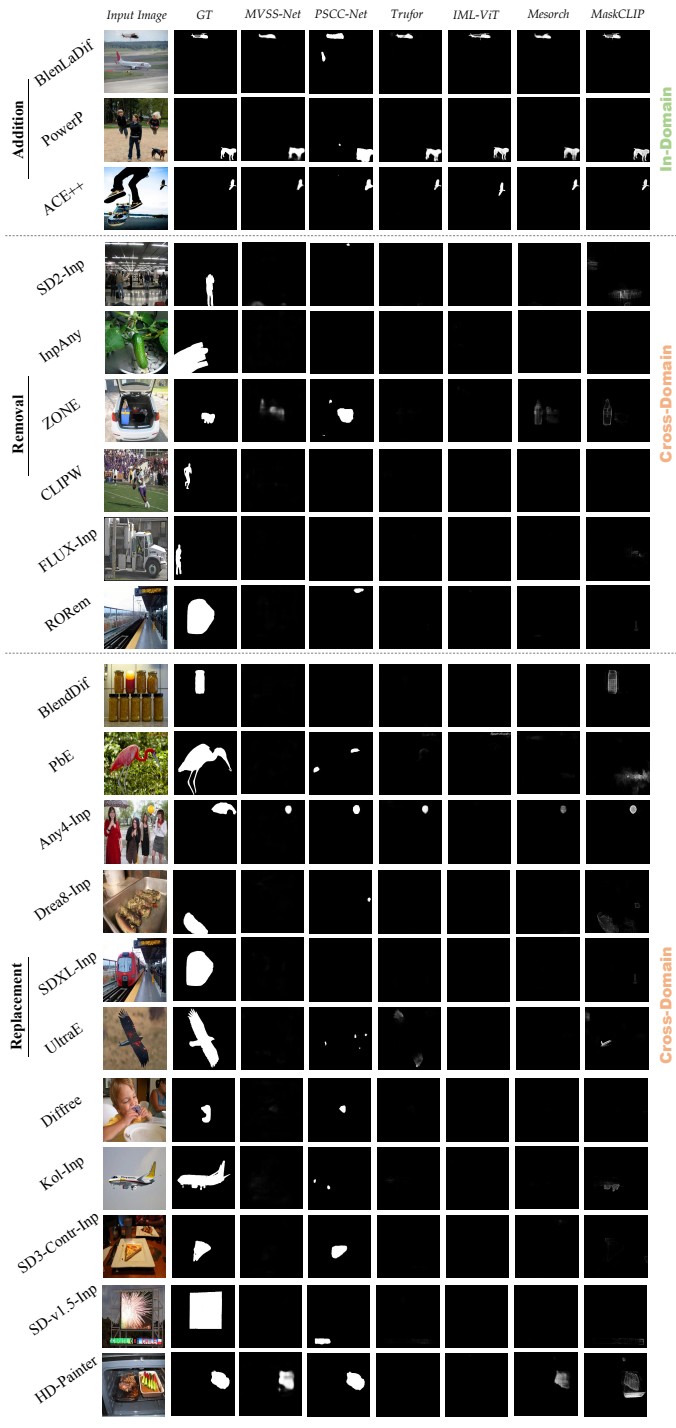

Figure 25: Visualization of models' prediction. **Protocol-2**. Trained on **Addition**.

## C.2 Models trained on replacement

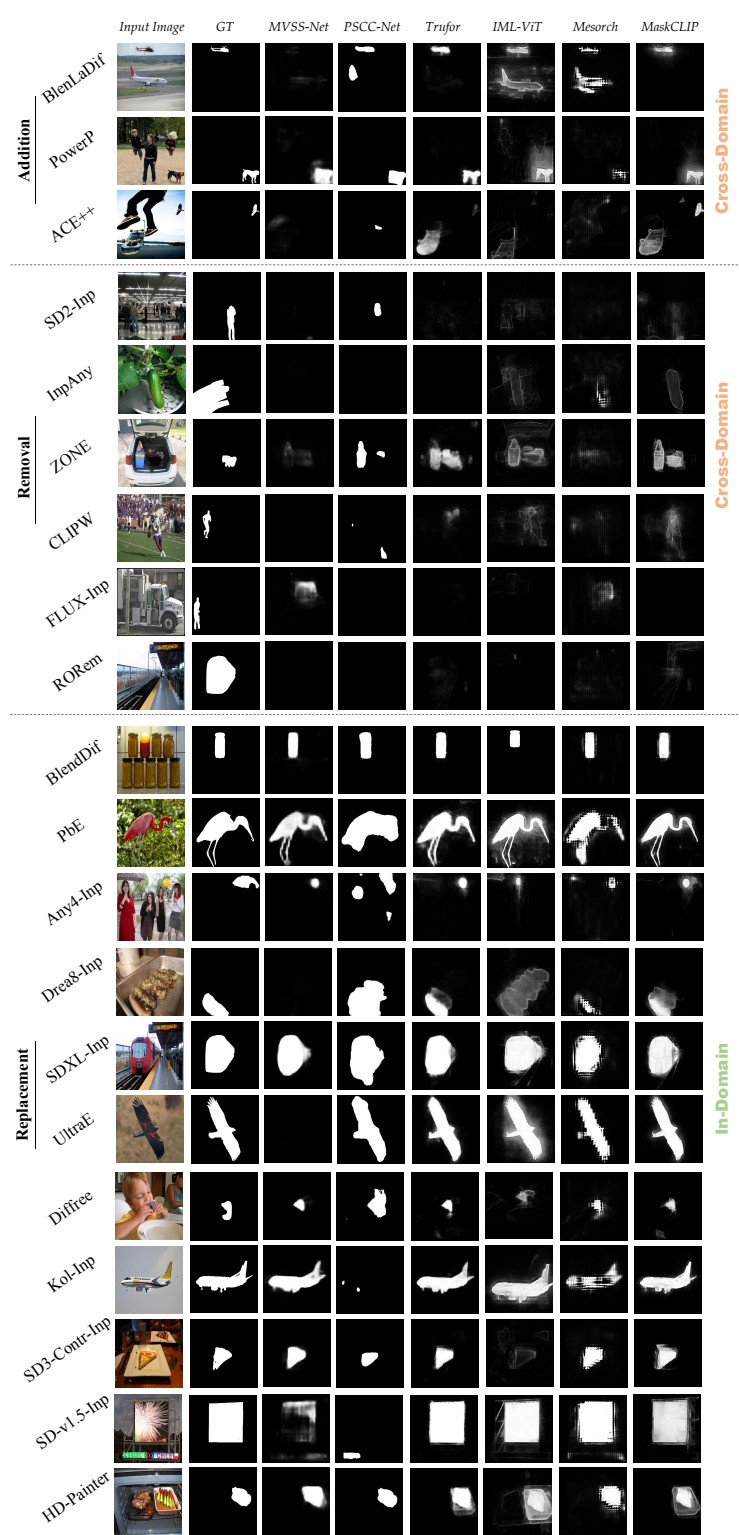

Figure 26: Visualization of models' prediction. **Protocol-2**. Trained on **Replacement**.

## D    USE OF LARGE LANGUAGE MODELS (LLMs)

During the preparation of this manuscript, we utilized Large Language Models (LLMs) as an assistive tool. The role of LLMs was strictly confined to improving the language and style of our writing, such as enhancing clarity, refining phrasing, and ensuring grammatical correctness for greater academic rigor. The use of LLMs did not extend to core aspects of the research, including ideation, analysis of experimental results, factual assertions, or the formulation of fundamental conclusions. All suggestions and text generated by LLMs were manually reviewed and edited by the authors to ensure the final text is accurate and faithfully represents our own work and intentions. The authors take full responsibility for all content presented in this paper.