# OpenReview forum: "NeXT-IMDL: Build Benchmark for NeXT-Generation Image Manipulation Detection & Localization"
_ICLR.cc/2026/Conference — ICLR 2026 Conference Withdrawn Submission_

### Official Review · Reviewer_bEAZ · 2025-10-20

**Soundness:** 2
**Presentation:** 1
**Contribution:** 2
**Rating:** 2
**Confidence:** 5

**Summary:**

The paper introduces NeXT-IMDL, a large-scale diagnostic benchmark for Image Manipulation Detection and Localization (IMDL) that systematically evaluates model generalization across AI-generated content. It covers 32 editing techniques (including GPT-Image-1 and Gemini-2.0), and defines five cross-dimension evaluation protocols based on editing models, manipulation types, semantics, and granularity. Experiments on 11 state-of-the-art detectors claim severe generalization failures under realistic settings, highlighting the need for more robust, next-generation IMDL models.

**Strengths:**

1. The authors strategically identify four critical failure modes that hinder IMDL generalization—cross-edit models, cross-edit types, cross-semantic labels, and cross-edit granularity. This formulation provides clear diagnostic dimensions and is experimentally well-supported, offering strong intuition for guiding future improvements in IMDL robustness.
2. The scale of the IMDL evaluation is impressive, encompassing 11 state-of-the-art models tested across 32 editing techniques, including recent tools such as GPT-Image-1 and Gemini-2.0.
3. Usage of four VLMs to reduce model-specific biases for proposal generation is thoughtful.
4. The ethics statement and reproducibility statements on page ten are very important for a task like IMDL benchmarking.

**Weaknesses:**

Major:
1. There are already comprehensive benchmark evaluations for IMDL in [1, 2] and for deepfake detection in [3], following a similar methodology to the proposed work. However, the authors neither compare their results with these prior benchmarks nor clarify how Next-IMDL improves upon IMDL-Benco [1]. Given the title Next-IMDL, it is reasonable to assume they are aware of the previous benchmarks, yet no direct comparison or analysis of improvements is presented. Is the inclusion of AIGC-based IMDL datasets the only distinguishing contribution? Why [2, 3] are not mentioned in related work?
2. The dataset generation pipeline described in the paper lacks sufficient detail for reproducibility. It is unclear how many proposal generation scenarios were used, how the automatic proposal generation process operates, and what specific edits or filters were applied. Figure 2 provides only a high-level overview and does not clarify these aspects. Are these implementation details—such as the proposal mechanisms, editing parameters, and filtering criteria—provided elsewhere in the main paper or the supplementary material?
3. Were all the state-of-the-art models trained solely on the NeXT-IMDL training split, or were there comparisons with the original pretrained weights of these models? The paper (lines 268–269) mentions that 558,269 images were divided into training, validation, and test sets in a 6:1:1 ratio, but it remains unclear whether the data distributions differ across these splits. Moreover, how did the authors ensure no overlap or data leakage within the dataset generation pipelines among the three subsets? Finally, Figure 2 illustrates the generation pipeline, but it is not specified whether it corresponds to the training, validation, or test set.
4. Discussions 1 and 2 (Lines 415–431) and Open Questions 1 and 2 (Lines 466–476) reference prior works but do not include experiments or analyses within this paper to substantiate those claims. Could the authors clarify whether any experiments or evaluations presented in this work directly support these discussions, or are these insights purely citation-based without empirical validation?
5. Can the authors experimentally verify that the observed performance degradation of state-of-the-art IMDL models on AIGC-manipulated images is indeed caused by the AIGC-specific manipulations, rather than confounding factors such as differences in model backbone architectures, pretraining strategies, or uneven evaluation settings among the prior SOTA methods? A controlled analysis isolating the effect of AIGC content from other architectural or training variations would strengthen the validity of the authors’ conclusions regarding model generalization failure.
6. Did the authors conduct any experimental evaluations on the quality and diversity of AIGC-generated images derived from different source datasets such as MSCOCO, Flickr30k, and OpenImages V7? While the paper claims to build a diverse benchmark using four different VLMs, it does not discuss whether the choice of source datasets introduces content or semantic biases into NeXT-IMDL. Were any analyses performed to assess dataset balance, domain bias, or source-dependent artifacts across these origins? Providing such evaluations would clarify the true diversity and representativeness of NeXT-IMDL and ensure that observed performance variations are not influenced by source dataset biases.



Minor:
1. Line 105 says three-fold contributions, but only two points of contributions are mentioned.
2. Tables 5, 6, and 7 are difficult to read and interpret due to limited caption details. The captions should be made more descriptive, clearly explaining the evaluation setup and defining all reported metrics. Additionally, the reference to Table 3.2.7 below Table 6 appears to be a numbering error and should be corrected for consistency.
3. Figure 1 (Lines 054–063) occupies considerable space merely to display the NeXT-IMDL symbol, offering limited informational value. This space could be more effectively utilized to present detailed insights into the dataset generation pipeline or to include additional qualitative and quantitative results that would enhance the paper’s substance and clarity.

[1] Ma et al., Imdl-benco: A comprehensive benchmark and codebase for image manipulation detection & localization, Neurips 2024

[2] Nandi et al. TrainFors: A large benchmark training dataset for image manipulation detection and localization, ICCV 2023

[3] Yan et al., Deepfakebench: A comprehensive benchmark of deepfake detection, Neurips 2023

**Questions:**

1. Did the authors conduct any experimental analysis to verify the loss of traditional artifact cues (e.g., those exploited by splicing, copy-move, and removal detection) when AIGC-based IMDL images are generated? Including quantitative evidence, such as frequency-domain plots or responses from SRM, Bayar, or DCT filters, could effectively demonstrate the degradation of these artifacts and strengthen the argument for developing new IMDL models tailored to AIGC-generated content.
2. Did the authors consider the issue of Vision-Language Model (VLM) hallucination during the proposal generation stage of the synthetic data pipeline for NeXT-IMDL? Since VLMs such as GPT-Image-1 and Gemini-2.0 can generate semantically inconsistent or visually incoherent edits that deviate from the input prompt, it is important to clarify how such hallucinated or low-quality generations were detected, filtered, or mitigated. Were there any manual verification, automatic quality control mechanisms, or semantic consistency checks (e.g., CLIP or BLIP-based alignment scores) applied to ensure the fidelity and realism of the generated manipulations? Without addressing VLM hallucination, the reliability and validity of the benchmark’s AIGC-generated samples could be compromised.
3. Did the authors perform any quality assurance or filtering on the images generated within the NeXT-IMDL synthetic data pipeline? Given that the dataset integrates outputs from multiple Vision-Language Models (VLMs) and image editing frameworks, it is important to clarify whether all generated samples were included in the training and evaluation sets, or if certain images were discarded due to low quality, unrealistic artifacts, or prompt inconsistency. Specifically, how were failed generations, over-edited samples, or visually implausible manipulations handled? Were there any objective filtering criteria (e.g., perceptual quality scores, CLIP similarity thresholds, or anomaly detection filters) or manual inspection protocols used to ensure that only high-quality, semantically aligned, and visually coherent images were retained? Providing these details would significantly enhance the transparency, reproducibility, and credibility of the dataset construction process.

**Details Of Ethics Concerns:**

While the authors acknowledge potential bias, fairness, and privacy issues, these are not empirically examined. The dataset sources (Flickr30k, MSCOCO, OpenImages V7) may introduce demographic and semantic biases, yet no quantitative bias analysis or filtering strategy is described. The automated generation pipeline and inclusion of human volunteers lack detail on privacy safeguards, consent documentation, and content screening for identifiable information. Additionally, the paper does not clarify access control or misuse prevention measures, raising security and dual-use concerns despite stated adherence to ethical guidelines.

---

### Official Review · Reviewer_g4QT · 2025-10-26

**Soundness:** 3
**Presentation:** 3
**Contribution:** 2
**Rating:** 4
**Confidence:** 5

**Summary:**

This paper introduces NeXT-IMDL, a large-scale diagnostic benchmark for IMDL. It systematically constructs a dataset along four axes (editing models, manipulation types, semantics, and granularity) and proposes five cross-dimension evaluation protocols to test model generalization. Extensive experiments on 11 models reveal significant performance drops under these challenging protocols, highlighting their fragility in real-world scenarios.

**Strengths:**

The benchmark is extensive, incorporating a wide variety of editing models (32) and covering multiple manipulation types and conditions. And the five cross-domain protocols are well-designed and effectively probe model weaknesses in generalization.

**Weaknesses:**

The work is primarily a dataset and evaluation framework. Its goal and methodology are very similar to prior benchmarks like IMDL-BenCo and GRE, making the conceptual advance somewhat limited. And the paper does not propose a new detection model or a novel theoretical insight; it is mainly a "stress test" for existing methods. Besides, some presentation issues exist, such as "Towars" in Figure 1 and the truncated label "Rem." in Line 111/Table 1, which detract from the overall polish.

**Questions:**

- How was the quality and difficulty consistency ensured across the diverse set of 32 editing models during dataset creation?
- Given the rapid evolution of generative models, how can NeXT-IMDL avoid becoming obsolete? Is the vision to create a living benchmark, perhaps with a community-driven platform for continuously integrating new editing models, rather than a static dataset release?
- The paper clearly demonstrates the systemic failures of current models. Beyond documenting the performance drops, do you have any deeper hypotheses on the underlying reasons? For instance, are models overfitting to specific generator "fingerprints," or are they failing to capture fundamental concepts of physical/lighting inconsistency that define a forgery?

---

### Official Review · Reviewer_Zp5q · 2025-10-26

**Soundness:** 3
**Presentation:** 3
**Contribution:** 4
**Rating:** 6
**Confidence:** 5

**Summary:**

This paper introduces NeXT-IMDL, a large-scale benchmark designed to systematically evaluate the generalization ability of IMDL models in the era of generative AI. The authors argue that current benchmarks create a benchmark illusion, where models perform well on limited in-distribution datasets but fail to generalize across unseen manipulation types, tools, and semantic contexts. To address this, NeXT-IMDL defines four orthogonal diagnostic axes to comprehensively probe model robustness. The benchmark includes 558K manipulated images generated using 32 editing models (both academic and commercial). Its systematic design, comprehensive dataset, and insightful analysis make it a strong benchmark contribution to the field.

**Strengths:**

1. The paper introduces NeXT-IMDL, a systematic benchmark that explicitly probes four key axes of IMDL generalization.
2. It provides a comprehensive dataset (558K samples) with 32 editing tools covering both academic and commercial generators.
3. The study delivers valuable empirical insights.
4. The experiments are extensive, involving representative IMDL models across five evaluation protocols.
5. The open-science commitment (planned dataset, code, and experiments release) enhances reproducibility and community impact.

**Weaknesses:**

1. The sections are dense, with complex cross-protocol discussions that may overwhelm general readers. Readability could be improved.

**Questions:**

1. Issues raised in the *Weaknesses* section.
2. Can NeXT-IMDL be extended to video manipulations?
3. Could the authors consider releasing a lightweight or partial subset of NeXT-IMDL for academic reproducibility?

---

### Official Review · Reviewer_ybkp · 2025-10-31

**Soundness:** 2
**Presentation:** 2
**Contribution:** 2
**Rating:** 2
**Confidence:** 4

**Summary:**

This paper looks at introducing a large scale dataset that includes a wide range of image editing techniques. They utilize a similar approach to OpenSDI whereby they utilize VLM’s for instructions, use SAM and LangSAM for creating the segmentations and edit the instructions, and then feed these into different image editing techniques. They then perform both detection and localization tests on their dataset. They propose 4 different tests such as Cross-Edit models, Cross-Edit types and so on to determine generalization weaknesses

**Strengths:**

* The dataset is a large scale dataset that includes a wide range of manipulation techniques across 4 axes of diversity with 3 different data sources.

**Weaknesses:**

* In Figure 1 the authors speak about evaluating on 5 protocols, however the Protocol 5: Toward-Realworld-IMDL does not appear to be present as a separate table in the main paper considering this is mentioned in the abstract and Figure 1, this seems very misleading if it is not present in the main paper. The point of Protocol 5 is to see the “lab-to-wild” performance, seeing how these models perform on commercial tools, the fact that this is not present is what I believe is a major oversight.

* The details of the volunteers are heavily obfuscated which seems a very important thing that needs to be more detailed, the authors say that a manual verification process takes places but does not detail the amount of people involved or how the manual verification is done, considering the size of the dataset I would want to know how the manual verification was done. Additionally in Figure 2 Human Labelling shows a figure of Photoshop and Gpt etc. but does not detail how this is used.

* What about using domain generalization techniques like SWAD, URM, MIRO [1,2,3] how does this affect performance, the dataset discusses how the models have systematic failures but do not go about trying accepted techniques like domain generalization to see the effect it has on the current models

[1] Swad: Domain generalization by seeking flat minima (Neurips 2021)

[2] Uniformly Distributed Feature Representations for Fair and Robust Learning (TMLR 2024)

[3] MIRO: Mutual Information Regularization with Oracle (ECCV'22)

* The authors utilize the “ground-truth” masks as where the manipulations take place; however many diffusion models have been shown to inpaint outside the mask, which raises the question of if the “ground truth” masks are the only inpainted regions [1]. Did you utilize a similar approach to OpenSDI where you only inpainted the region that contained the mask?

[1] OpenSDI: Spotting Diffusion-Generated Images in the Open World [CVPR 2025]

* Additionally the paper does not look at how the models perform when trained on their proposed dataset and then are evaluated on datasets that utilize diffusion based inpaintings like OpenSDI, GRE, SIDA and DOLOS, CocoGLIDE, MagicBrush to evaluate on [1,2,3,4]. Also how do the models perform on traditional manipulations like CASIA, TGIF, IMD2020 to name a few.

[1] OpenSDI: Spotting Diffusion-Generated Images in the Open World [CVPR 2025)

[2] Rethinking Image Editing Detection in the Era of Generative AI Revolution (ACM MM, 2024)

[3] Sida: Social media image deepfake detection, localization and explanation with large multimodal model. (CVPR 2025)

[4] Weakly-supervised deepfake localization in diffusion-generated images (WACV 2024)

[5] Trufor: Leveraging all-round clues for trustworthy image forgery detection and localization (CVPR 2023)

[6] Magicbrush: A manually annotated dataset for instruction-guided image editing (NEURIPS 2023)




* None of the tables captions really have any description of what the table contains, for instance the metrics are not mentioned in Table 3 and do not describe what the distinction between Set X and Set y on the row/column mean, does it mean trained on Set Y and tested on Set X, this is unclear.

* Because of the composition of the dataset having numerous image editing techniques it can be harder to determine if the reason methods like BlendedDiffusion and Anything-4.0-Inpainting performs worse is because of the smaller amount of images or because of the actual manipulation technique.

* Currently the way the table of results are reported it can be very difficult to determine the average across the different sets, adding an average for a set would be useful.

* There does not appear to be much distinction between the different images sources (e.g. Flickr30K versus COCO) which is expected to be a major influence on the results as is common in domain generalization scenarios [1,2,3]. It is unclear how the image sources are affecting the results.

[1] Swad: Domain generalization by seeking flat minima (Neurips 2021)

[2] Uniformly Distributed Feature Representations for Fair and Robust Learning (TMLR 2024)

[3] MIRO: Mutual Information Regularization with Oracle (ECCV'22)

* Table 3 appears to have repeated values for the rows Set 1 and Set 2 which raises concerns about the validity of the results of this paper as such a glaring large set of incorrect numbers were overlooked

**Questions:**

* Where are the results and discussion of protocol 5 and why is it not in the main paper since it is mentioned in the abstract and figure 1.
* More detail needs to be done on the human evaluation
* Why were no domain generalization techniques considered to try to bridge the gap of issues of generalization
* The different tables do not describe what the different sets mean, I assume the row and columns sets mean you are trained and then tested on different sets, but there is barely any captions used in the tables.
* For Table 3, why are the results for trained on set 1 and test on set 1 the same as train on set 2 and test on set 2? This seems very concerning since for Table 3 Set 1 and Set 2 for the rows, have the same exact rows, additionally the second best value for Set 1 is reported as 0.656 but then 0.624 is reported as the best for Set 2. This is deeply concerning as it appears that values were just repeated.

---

### Note · Authors · 2025-11-26

I have read and agree with the venue's withdrawal policy on behalf of myself and my co-authors.